# Targeting Inflammation with Natural Products: A Mechanistic Review of Iridoids from Bulgarian Medicinal Plants

**DOI:** 10.3390/molecules30173456

**Published:** 2025-08-22

**Authors:** Rositsa Mihaylova, Viktoria Elincheva, Reneta Gevrenova, Dimitrina Zheleva-Dimitrova, Georgi Momekov, Rumyana Simeonova

**Affiliations:** 1Department of Pharmacology, Pharmacotherapy and Toxicology, Faculty of Pharmacy, Medical University of Sofia, 1000 Sofia, Bulgaria; v.lyubomirova@pharmfac.mu-sofia.bg (V.E.); gmomekov@pharmfac.mu-sofia.bg (G.M.); 2Department of Pharmacognosy, Faculty of Pharmacy, Medical University of Sofia, 1000 Sofia, Bulgaria; rgevrenova@pharmfac.mu-sofia.bg (R.G.); dzheleva@pharmfac.mu-sofia.bg (D.Z.-D.)

**Keywords:** iridoids, inflammation, antioxidant, metabolic syndrome, NFkB, JAK/STAT, AMPK, catalpol, aucubin, non-communicable diseases

## Abstract

Chronic, low-grade inflammation is a key contributor to the development of numerous non-communicable diseases (NCDs), including cardiovascular, metabolic, and neurodegenerative disorders. Conventional anti-inflammatory drugs, such as nonsteroidal anti-inflammatory drugs (NSAIDs) and corticosteroids, often present safety concerns with prolonged use, highlighting the need for safer, multi-targeted therapeutic options. Iridoids, a class of monoterpenoid compounds abundant in several medicinal plants, have emerged as promising bioactive agents with diverse pharmacological properties. They exert anti-inflammatory and metabolic regulatory effects by modulating key signaling pathways, including nuclear factor kappa B (NF-κB), mitogen-activated protein kinase (MAPK), Janus kinase/signal transducer and activator of transcription (JAK/STAT), adenosine monophosphate-activated protein kinase (AMPK), and peroxisome proliferator-activated receptor (PPAR) pathways. This review provides a comprehensive summary of the major iridoid metabolites derived from ten Bulgarian medicinal plant species, along with mechanistic insights from in vitro and in vivo studies. Documented biological activities include anti-inflammatory, antioxidant, immunomodulatory, antifibrotic, organoprotective, antibacterial, antiviral, analgesic, and metabolic effects. By exploring their phytochemical profiles and pharmacodynamics, we underscore the therapeutic potential of iridoid-rich Bulgarian flora in managing inflammation-related and metabolic diseases. These findings support the relevance of iridoids as complementary or alternative agents to conventional therapies and highlight the need for further translational and clinical research.

## 1. Introduction

Inflammation is a tightly regulated physiological process that plays a central role in the body’s defense against infection, injury, and other noxious stimuli aimed at restoring internal homeostasis. Although acute inflammation is essential for healing and host defense, persistent, chronic inflammation is increasingly recognized as a key driver in the pathogenesis of numerous non-communicable diseases (NCDs), including cancer, cardiovascular diseases, neurodegenerative disorders, type 2 diabetes, obesity, metabolic syndrome, and autoimmune conditions [1]. While genetic, environmental, and lifestyle factors are major contributors to the pathogenesis of NCDs, chronic low-grade inflammation emerges as a shared and critical mechanistic link in their complex interplay [2].

Inflammatory responses are orchestrated by conserved signaling pathways such as NF-κB, MAPK, and JAK/STAT, which regulate cytokine production, immune cell activation and differentiation, cell survival, and reactive oxygen species (ROS) formation. In addition, the AMP-activated protein kinase (AMPK) signaling acts as a central energy sensor that couples metabolic regulation with anti-inflammatory and antioxidant defenses. When cellular energy is depleted, AMPK activation shifts metabolism toward catabolic pathways and restores energy balance, while suppressing lipogenesis and inflammatory signaling associated with metabolic syndrome [3]. Importantly, these pathways are often engaged in dynamic crosstalk interactions, further amplifying the inflammatory signals by initiating and sustaining pathological positive feedback loops [4]. 

Given the multifaceted nature of these signaling pathways, there is a growing demand for therapeutic agents that can target inflammation in a multi-modal and balanced manner, with fewer side effects than traditional NSAIDs or corticosteroid drugs.

Within the inflammatory signaling network, natural products, particularly plant-derived phytochemicals, have emerged as promising candidates, owing to their structural diversity, multitarget potential, and more favorable safety profiles. Much of the research to date has focused on the anti-inflammatory activity of well-known compounds such as curcumin, resveratrol, or quercetin [5,6], while a wide array of understudied secondary metabolites and their plant sources remain understudied or lack in-depth evaluation in the context of complex inflammatory diseases and their molecular mechanisms.

Indeed, polyphenols are among the most extensively studied and structurally diverse secondary metabolites, valued for their pleiotropic anti-inflammatory, antioxidant, and immunomodulatory activities. Flavonoids, the most abundant class of polyphenols, exert potent anti-inflammatory effects through modulation of nuclear factor kappa B (NF-κB), mitogen-activated protein kinase (MAPK), and signal transducer and activator of transcription (STAT) pathways, and reducing the expression of pro-inflammatory enzymes such as cyclooxygenase-2 (COX-2), inducible nitric oxide synthase (iNOS), and lipoxygenases (LOXs). Chlorogenic acid, a widely distributed acylquinic acid, suppresses pro-inflammatory mediators including TNF-α, IL-1β, IL-6, IL-8, NO, and PGE_2_, thereby providing cellular and tissue protection against various inflammatory conditions [7,8].

Like polyphenols, iridoids represent another important group of plant secondary metabolites with notable anti-inflammatory potential, owing to their complex modulatory activity on key signaling pathways involved in immune and oxidative stress responses, including nuclear factor erythroid 2-related factor 2 (Nrf2), nuclear factor kappa B (NF-κB), mitogen-activated protein kinase (MAPK), Janus kinase/signal transducer and activator of transcription (JAK/STAT), adenosine monophosphate-activated protein kinase (AMPK), and peroxisome proliferator-activated receptor (PPAR) signaling [9,10]. This shared ability of structurally distinct metabolite classes to target overlapping inflammatory pathways, often acting synergistically, underscores the broad therapeutic promise of plant-derived compounds as complementary or alternative anti-inflammatory agents.

However, iridoids further demonstrate unique mechanistic properties. While certain flavonoids and polyphenols have been shown to enhance endogenous glucagon-like peptide-1 (GLP-1) secretion or inhibit its degradation (e.g., via DPP-4 inhibition), iridoids appear to go a step further: emerging preclinical evidence indicates that several iridoid glycosides, such as geniposide, shanzhiside methylester, 8-O-acetyl shanzhiside methyl ester (8-OaS), morroniside, and catalpol, may act as direct small-molecule agonists of the GLP-1 receptor (GLP-1R). This receptor plays a central role in glucose regulation, insulin secretion, and appetite control. Through this receptor-level interaction, iridoids offer promising therapeutic potential for metabolic conditions such as type 2 diabetes, obesity, non-alcoholic fatty liver disease (NAFLD), hypertension, and cardiovascular disorders, many of which are tightly linked to chronic inflammation and metabolic dysfunction [11,12].

The purpose of the present study is to provide a comprehensive review, addressing the molecular mechanisms of plant-derived iridoids, with a particular focus on their interactions with key molecular targets and signaling pathways involved in inflammation. Within the scope of our research, we focus on the pharmacodynamic profile of iridoid secondary metabolites found in selected medicinal plant species native to the Bulgarian flora. Furthermore, we summarize existing in vitro and in vivo evidence supporting the therapeutic potential of these plants as both plant-based remedies and a source of iridoid glycosides as principal bioactive compounds. By elucidating their mechanisms of action and pharmacological effects, we seek to emphasize the potential of iridoid derivatives and their plant sources as valuable alternatives to conventional anti-inflammatory therapies.

## 2. Bulgarian Species Rich in Iridoids: Phytochemistry and Therapeutic Potential

Iridoids represent a diverse class of monoterpenoid secondary metabolites that are widespread across the plant kingdom, particularly among families such as Plantaginaceae, Lamiaceae, Scrophulariaceae, and Rubiaceae—all of which are well-represented in the Bulgarian flora [13]. These compounds are biosynthetically derived from geraniol as a precursor in a biosynthetic pathway, different than the one producing monoterpenes commonly found in essential oils, leading to the formation of a characteristic cyclopenta[c]pyran scaffold [14]. The bicyclic cyclopenta[c]pyran skeleton serves as a key pharmacophore with distinct structural and functional features that contribute to both the ecological roles of iridoids in plant defense and the therapeutic properties of the medicinal species [15]. 

Structurally, iridoids typically occur as glycosides (iridoid glycosides) and less commonly as aglycone forms, secoiridoids (resulting from the enzymatic cleavage of the 7,8-cyclopentane bond), and bisiridoids dimers formed through the coupling of two iridoid or iridoid-secoiridoid units. The structural versatility of the cyclopenta[c]pyran core enables diverse functional modifications, resulting in compounds with distinct bioactivities [16]. The present review is focused on elucidating the mechanistic aspects of iridoid glycosides found in species native to Bulgaria, traditionally used for their hepatoprotective, neuroprotective, antidiabetic, and general anti-inflammatory properties. Among these, aucubin and catalpol, sharing high structural resemblance (differing by the presence of an epoxide group in catalpol’s structure), are the most widely distributed and well-studied iridoid glycosides, occurring in multiple native to Bulgaria species such as *Veronica officinalis*, *Plantago* spp., *Verbena officinalis*, *Verbascum phlomoides,* and *Gratiola officinalis* [17,18,19]. Furthermore, we have explored the pharmacodynamic profiles of several iridoid metabolites with a more species-specific distribution, such as verbenalin and hastatoside (*Verbena officinalis*), harpagoside (*Scrophularia nodosa*, *Gratiola officinalis*), lamalbid, asperuloside, and monotropein (*Galium* spp.), shanzhiside, and its 8-acetyl methylester (8-OaS) (*Lamium album*) (Figure 1).

### 2.1. Veronica officinalis

*Veronica officinalis*, a member of the Plantaginaceae family, is a species widely distributed across Europe and traditionally employed in folk medicine to treat stomach and intestinal disorders and colic, renal lithiasis, pulmonary diseases, and skin wounds [17,20]. Phytochemical profiling of *V. officinalis* reveals a rich composition of bioactive constituents, including iridoid glucosides (aucubin, catalpol, verproside), phenolic acids (p-coumaric, ferulic acids), flavonoids (luteolin, apigenin, quercitrin, quercetin), and sterols (β-sitosterol) [20].

Among these compounds, verproside (catalpol derivative iridoid glycoside) has been recognized as a dominant active constituent of the species and has been extensively studied for its therapeutic potential. Standardized extracts of *V. officinalis* were found to inhibit pro-inflammatory mediators in human lung epithelial cells (A549) by reducing COX-2 expression and PGE2 synthesis. Additionally, the treatment with the extract diminished eotaxin expression, a chemokine pivotal for inflammatory cell recruitment. Accordingly, verminoside and verproside were identified as key iridoid glycosides, accountable for the efficacy of these extracts [20].

In vivo, *V. officinalis* preparations demonstrated anti-ulcerogenic properties in indomethacin-induced ulcer models and enhanced mucosal regeneration in reserpine-induced gastric injury in rats [21]. The hepatoprotective activity of the species was demonstrated in CCl_4_-induced hepatic injury, where extracts normalized antioxidant enzyme activities (catalase, peroxidase) and glutathione levels while reducing lipid peroxidation markers [22]. Additional studies monitoring malondialdehyde and glutathione in rats corroborate its protective effect against oxidative stress [23].

In a placebo-controlled clinical trial, topical application of a formulated cream containing *Veronica officinalis* extract (Scoti-Speedwell™) for 56 days resulted in a significant reduction in wrinkle area by 18.0% and wrinkle length by 16.05% in female subjects. In vitro assays demonstrated that the ethanolic extract exhibited notable DPPH free radical scavenging activity and significantly enhanced collagen synthesis, supporting the skin-healing properties and therapeutic potential of the plant as an effective anti-wrinkle agent in human skin [24].

Other *Veronica* species also exhibit similar biological profiles: *V. urticifolia* and *V. jacquinii* have been ascribed potent COX-1, 12-lipoxygenase inhibitory activity, both critical in modulating the arachidonic acid inflammatory cascade [25]. Furthermore, aqueous fractions of several *Veronica* species demonstrated significant antioxidant properties, due to radical scavenging and decreased NO production in macrophages [26].

### 2.2. Plantago spp.

*Plantago major* and *Plantago lanceolata*, both perennial species from the Plantaginaceae family, are globally distributed and have been traditionally valued for their antimicrobial, antidiabetic, antispasmodic, antiviral, anti-inflammatory, and wound-healing properties [27,28].

*Plantago major* is notably rich in phenolic compounds and flavonoids, including luteolin, apigenin, baicalein, hispidulin, plantagin, and scutellarin, alongside terpenoids and iridoid glycosides such as aucubin, asperuloside, majoroside, catalpol, gardoside, and melittoside. It also contains caffeic acid derivatives like plantamajoside and acteoside (often referred to as verbascoside) [27,28,29]. *Plantago lanceolata* shares a similar phytochemical profile, being a rich source of aucubin, catalpol, acteoside, asperuloside, globularin, isoverbascoside, plantamajoside, and lavandufolioside [28].

The anti-inflammatory potential of *Plantago* species has been demonstrated by numerous experimental studies. Water and ethanol extracts of *P. major* leaves showed significant anti-inflammatory effects in oral epithelial cells in vitro [30]. In vivo studies using a rat model of acetic acid-induced ulcerative colitis revealed that *P. major* leaf extracts markedly reduced the ulcerative index, histopathological changes, and levels of inflammatory markers such as IL-6, TNF-α, PGE2, IL-1β, MPO, and MDA [31]. Additionally, microemulsified ethanolic extracts of the plant effectively attenuated Croton oil-induced ear edema in mice, with efficacy comparable to 1% hydrocortisone [27]. Further pharmacological evaluations demonstrated that both soluble and insoluble fractions of a dichloromethane-derived *P. major* extract confer notable anti-inflammatory activity. Specifically, these extracts inhibited COX-2–mediated prostaglandin biosynthesis and reduced leukocyte infiltration in several animal models of inflammation, including thioglycolate-induced peritonitis and paw edema. In a model of rheumatoid arthritis, treatment with the more active isobutanol fraction suppressed osteoclast development and COX-2 expression, an effect linked to the presence of fatty acids such as oleic and linoleic acids, known for their ability to modulate inflammatory mediators production [32].

These promising in vivo findings for *Plantago major* L. have also been translated into clinical settings, where the plant has been evaluated for various inflammatory and tissue-repair indications. Notably, its traditional wound-healing properties have been substantiated in multiple randomized trials. A triple-blind study in patients with stage 1 pressure ulcers demonstrated that a standardized *P. major* topical formulation significantly accelerated lesion resolution without adverse effects, validating its ethnomedicinal use [33]. Similarly, an open-label trial in diabetic foot and pressure ulcer patients reported that 10% *P. major* hydroalcoholic gel promoted greater wound size reduction than control dressings at both week 1 and week 2, with a markedly higher complete healing rate (64% vs. 20.45%) [34]. Beyond wound repair, clinical evidence has also suggested a broader anti-inflammatory potential of the plant. In ulcerative colitis patients, *P. major* seed supplementation for 8 weeks alleviated abdominal tenderness, reflux, gastric pain, and reduced visible blood in stool [35]. In oncology care, a randomized, double-blind trial in head and neck cancer patients showed that *P. major* syrup significantly reduced the severity of radiation-induced oral mucositis and pain [36]. Earlier reports in chronic bronchitis patients also described rapid symptom improvement, better respiratory indices, and excellent tolerability [37].

Similar to *Plantago major*, *Plantago lanceolata* L. has been traditionally used to treat skin inflammations and respiratory ailments. Its leaf extracts, rich in ursolic and oleanolic acids, showed potent topical anti-inflammatory effects in vivo, with ursolic acid being more effective than indomethacin [38]. In the context of infectious inflammation, *P. lanceolata* also exhibited notable antibacterial and biofilm-inhibiting activities against *Borrelia burgdorferi*, effects partially attributed to plantamajoside, acteoside, and volatile compounds such as β-caryophyllene [28,39].

Several pharmaceutical preparations derived from *Plantago lanceolata* (e.g., syrups, lozenges, liquid and solid extracts) are in established clinical use, particularly for the treatment of dry cough and inflammation of the upper respiratory tract. Additionally, these formulations are applied in skin regeneration, toothache relief, and as antibacterial agents, as well as for supporting immune function [40]. In Europe, folium *Plantaginis* extracts are used as single plant or polyherbal preparations for various medicinal purposes, including digestion problems (Finland, Romania), as expectorant (Slovenia, Italy, Romania, Bulgaria), antimicrobial, astringent, and irritation soothing remedies (Poland, Belgium) [41].

### 2.3. Verbena officinalis

*Verbena officinalis* is a well-known medicinal plant, distributed globally—in Europe, the Americas, North and Central Africa, Asia, and Australia, and used for centuries as traditional medicine to treat rheumatism, bronchitis, depression, insomnia, anxiety, liver, and gallbladder diseases [42,43]. The herb is a rich source of bioactive compounds, including iridoid glycosides (verbenalin, aucubin, hastatoside), phenylpropanoid glycosides (verbascoside, isoverbascoside, eukovoside), flavonoids (luteolin, apigenin, kaempferol, scutellarein, pedalitin), phenolic acids (chlorogenic, ferulic, rosmarinic acids), and terpenoids [42,44].

Semi-polar to polar solvent extracts of *Verbena officinalis* (rich in iridoids and polyphenols) exhibited the highest antioxidant capacity in vitro, as demonstrated by their strong performance in DPPH, ABTS, CUPRAC, FRAP, and PMA assays [45,46]. Verbascoside and luteolin glucuronides were identified as key contributors to these effects, with luteolin 7-O-(2″-glucuronyl)-glucuronide demonstrating high binding affinities in molecular docking studies targeting proteins involved in oxidative stress and metabolic regulation [46]. Furthermore, bioactivity assays revealed significant enzyme inhibitory properties, including urease (comparable to hydroxyurea), acetylcholinesterase, butyrylcholinesterase, tyrosinase, α-amylase, and α-glucosidase inhibition [45,46].

Consistently, these anti-inflammatory effects have also been validated by in vivo studies. For example, methanolic and chloroform extracts significantly reduced carrageenan-induced paw edema in rodents, with efficacy comparable to piroxicam [47]. Topical formulations containing 3% extract alleviated inflammation and pain, though less effectively than methyl salicylate, with iridoids, caffeoyl derivatives, and flavonoids considered as major functional components [14].

Immunomodulatory and antiviral activities have also been reported. Oral administration of the extract in a murine influenza model mitigated lung injury, promoted NK cell maturation and activation, and reduced serum IL-6, TNF-α, and IL-1β levels [43]. Verbenalin specifically enhanced NK cell cytotoxicity by accelerating killing kinetics without affecting proliferation or degranulation. Lastly, network pharmacology and molecular docking studies identified quercetin, luteolin, and kaempferol as principal agents targeting AKT1, IL-6, and TNF-α, pointing to the therapeutic potential of *V. officinalis* in atherosclerosis and other systemic inflammatory diseases [48].

To date, clinical data on the therapeutic efficacy of *Verbena officinalis* L. is scarce. However, a decoction of the plant has been studied in a double-blind, randomized multicenter trial of 260 patients with chronic generalized gingivitis, where it achieved a significant reduction in Gingival Index and Plaque Index over 28 days compared to standard oral hygiene alone. No adverse effects were reported, indicating the decoction is a safe and effective approach for gingivitis management [44]. In addition, toxicological studies indicate that various *Verbena* herb extracts are low-toxic or virtually non-toxic when administered orally. Combined with its antimicrobial, anti-inflammatory, and antioxidant properties, this high safety profile supports the potential use of the species in treating respiratory system and other inflammatory conditions [49].

### 2.4. Scrophularia nodosa

*Scrophularia nodosa*, commonly known as Figwort, is a herbaceous plant, primarily distributed in temperate Eurasia and the Northern Hemisphere. This species is noted for its richness in bioactive compounds, particularly iridoid glycosides, as well as phenylpropanoids, phenolic acids, flavonoids, saponins, and cardiac glycosides. These constituents underlie its broad spectrum of traditional therapeutic properties, including anti-inflammatory, immune-modulating, antibacterial, hepatoprotective, cardioprotective, diuretic, protozoocidal, fungicidal, and antitumour activities [50,51,52].

Several iridoids have been isolated from the species, including aucuboside, catalpol, catalposide, and ester derivatives of harpagide, alongside phenolic compounds such as ferulic acid [50]. These compounds contribute significantly to the anti-inflammatory, analgesic, and antioxidant properties of the plant, as evidenced by mechanistic studies [51].

Beyond anti-inflammatory actions, iridoids in *Scrophularia* species, including *S. nodosa*, have shown potential immunomodulatory effects. Although these effects have not been directly studied in *S. nodosa*, structurally analogous to catalpol iridoids (e.g., scropoliosides) from related species have demonstrated various therapeutic activities, including hepatoprotective, immunomodulating, and anti-inflammatory, mostly mediated through NF-kB and inflammasome NLRP3 pathway inhibition, verifying that the catalpol nucleus common to many iridoids may be integral to their biological properties [50,53,54]. While no clinical data is available for *Scrophularia nodosa*, a related iridoid-rich species (*S. striata*) showed efficacy in a randomized trial for chronic periodontitis, outperforming standard mouthwash in improving plaque index, pocket depth, and bleeding on probing [55].

### 2.5. Verbascum phlomoides

*Verbascum phlomoides* is a member of the Scrophulariaceae family, traditionally employed across Eurasia for respiratory ailments such as bronchitis, asthma, and spasmodic cough [56,57]. The species is reported to be particularly rich in iridoid glycosides (e.g., aucubin, catalpol, specioside), flavonoids (including apigenin, luteolin, kaempferol, quercetin derivatives), phenylethanoid glycosides (verbascoside, verproside), triterpene saponins, and phenolic acids (e.g., caffeic, ferulic, protocatechuic acids [19,58].

Current research on the therapeutic potential of *Verbascum phlomoides* extracts remains limited and underexplored. A polyphenol-rich aqueous/butanol extract of dried *V. phlomoides* flowers has been reported to exhibit antioxidant properties and to inhibit TNF-α-induced ICAM-1 expression in endothelial cells by nearly 60% [56,58]. Ethanol extracts of *V. phlomoides* also displayed significant acetylcholinesterase and tyrosinase inhibition, linked to phenolic content and suggesting potential use for neuroprotection [58].

However, verbascoside (also named acteoside) and aucubin, identified as the primary phytoconstituents responsible for the species’ anti-inflammatory activity, have been more thoroughly studied and appear to act in a synergistic manner. Verbascoside is a water-soluble phenylethanoid glycoside, exhibiting a broad spectrum of bioactivities, including antioxidant, neuroprotective, immunomodulating, and anti-microbial, and has undergone extensive in vitro and in vivo investigation [59]. More importantly, it has been clinically evaluated in various inflammatory conditions, including COVID-19, where it contributed to cytokine regulation and reduction in systemic inflammation [59]. Ongoing research aims to improve its bioavailability through advanced delivery systems to support its clinical translation [60].

### 2.6. Galium spp.

*Gallium verum*, *G*. *aparine,* and *G. odoratum* (Rubiaceae family) are natively distributed throughout Eurasia and northern Africa. They typically produce carbocyclic iridoids; however, only in *G. verum* have iridoid esters with p-hydroxyphenylpropionic acid, like the iridoid V1, been found [16,61]. Loganin and 6-acetylscandoside are also characteristic of the *G. verum* species. The iridoid profiles of *G. aparine* and *G. odoratum* were dominated by geniposidic acid, 10-deacetylasperulosidic acid, scandoside, monotropein, asperulosidic acid, deacetylasperuloside, and asperuloside [61,62]. 

Asperuloside is the most consistently found iridoid glycoside across all species of the genus *Galium*, with *G. verum* and *G. mollugo* being the richest sources. Other iridoid derivatives include loganin, geniposidic acid, and scandoside, and contribute to the immunomodulatory, hepatoprotective, neuroprotective, anti-inflammatory, and antibacterial properties of the genus [63].

Studies revealed that the *G. verum* methanolic extract alleviated cardiac oxidative stress and decreased the generation of pro-oxidants [64,65]. It is worth noting that it prevented the pathological injuries caused by doxorubicin. Intake of *G. verum* extract was associated with a reduction in most of the measured pro-oxidants (NO_2_-, O_2_-, and TBARS levels) in comparison to the DOX group. Moreover, this extract was capable of increasing SOD and CAT activities [64].

In an in vitro endothelial model, *G. verum* extract has demonstrated antiangiogenic and antioxidant properties and exerted strong anti-inflammatory activity by significantly reducing the levels of pro-inflammatory cytokines IL-8 and IL-6. [66]. A methanol–aqueous extract of *G. verum* has also shown great potential in managing gastric injury in an in vivo model of an ethanol-induced gastric ulcer in rats. The plant extract exerted a significant decrease in ulcer index, gastric juice volume, malondialdehyde, and nitric oxide, while gastric juice pH, glutathione, glutathione-S-transferase, and catalase increased significantly. The histological examination confirmed the gastroprotective effects. Anti-inflammation activity was evidenced to be mediated by the down-regulation of the markers NF-κB p65 and TNF-α in the *G. verum* extract-treated group. This study highlights [67] that the immuno-enhancing effects of *Galium aparine* aqueous extract have been studied in cyclophosphamide-induced immunodeficient animals [68]. Gene expression analysis displayed differences in the individual gene expression levels of a series of genes, such as Entpd1, Pgf, Thdb, Syt7, Sqor, and Rsc1al in the *G. aparine* extract-treated group. In another study, *G. odoratum* methanolic and aqueous extracts exhibited anti-inflammatory activity, effectively reducing the pro-inflammatory M1 phenotype macrophage population involved in inflammatory responses [69].

### 2.7. Nepeta cataria L.

*Nepeta cataria*, catnip or catmint, is a perennial herbaceous plant, native to southern and eastern Europe, northern parts of the Middle East, and Central Asia. The plant species is famous for its euphoric effect on domestic cats and possesses valuable secondary metabolites with medicinal properties. Catnip has been used in infusions, tinctures, teas, juices, and poultices; due to its lemony mint flavor, the plant has been frequently used in cooking as well. *N. cataria* has been widely used in traditional medicine as a folk remedy for cough and cold, colic, asthma, and diarrhea. Catnip is of scientific interest largely due to the production of nepetalactones, volatile iridoid terpenes with strong arthropod repellent activity. In addition, the plant can also produce other bioactive volatile iridoids, such as nepetalic acid, nepetalactam, and dihydronepetalactone [70]. The anti-inflammatory activity of *N. cataria* extract from flowers, upper and lower leaves, was evaluated by measuring nitric oxide concentrations after treating RAW 264.7 murine macrophages stimulated by *E. coli* lipopolysaccharide. All tested extracts demonstrated radical scavenging capabilities and dose-dependent activity in the nitric oxide [71]. *N. cataria var. citriodora* essential oil, rich in nepetalactone E,E- and E,Z- nepetalactone isomers, demonstrated peripheral anti-inflammatory properties by reducing the induced edema after carrageenan injection. The tail immersion test suggested that the pharmacological actions were mediated by μ-opioid receptors rather than by k- and *δ*-receptors. Moreover, the peripheral analgesic action of the oil on acetic acid-induced pain was comparable to that of morphine, suggesting that the primary action of *N. cataria* essential oil involves central nervous system pathways associated with pain modulation [72].

### 2.8. Lamium album L.

*Lamium album* L., also known as white dead-nettle, inhabits Europe and Asia. *L. album* exhibits a wide range of therapeutic properties, which are assigned to various biologically active phytocompounds, including iridoids, phenolic acids, flavonoids, isoscutellarein derivatives, terpene, and essential oil phytoecdysteroids. Iridoid glycosides, such as lamiridoside (lamalbid), lamiol, lamiridozin A and B, caryoptoside, shanzhiside, and its 8-acetyl methyl ester (8-OaS), dominate the phytochemical profile and contribute to the broad anti-inflammatory, antibacterial, antioxidant, antiviral, immunomodulatory, as well as wound-healing properties of the plant [29,73,74,75], and the secoiridoids albosides A and B (albosides A and B) have also been identified [76].

*L. album* extracts from flowers and aerial parts have been found to inhibit the production of inflammatory mediators like cytokines (IL-8 and TNF-α), reactive oxygen species (ROS), and myeloperoxidase secretion in human neutrophils [77]. Methanol, ethyl acetate, and heptane extracts of *L. album* were investigated for their ability to stimulate the growth of human skin fibroblasts (HSF) in vitro. The heptane extract showed a favorable toxicity profile in human skin fibroblasts, even at high concentrations, and exerted stimulatory effects on HSF cells viability and proliferation [78]. An in vivo study revealed the potential anti-inflammatory effect of *L. album* extract through inhibition of caspase-3 and cyclooxygenase-2 gene expression in a rat model of middle cerebral artery occlusion (MCAO). The mRNA expression of caspase-3 in the core, penumbra, and subcortex regions in the extract-treated group was significantly reduced compared to the unprotected MCAO animals. Furthermore, the expression level of the inducible COX-2 isoform in the subcortex of the rats was decreased in response to treatment, as well as infarct size in the core, penumbra, and subcortex [79]. Similarly, in streptozotocin-induced diabetic rats, *L. album* aerial parts extract inhibited the liver and kidney expression of both COX-2 and caspase-3, mitigating organ damage and attenuating inflammation-induced apoptotic cell death [79].

### 2.9. Euphrasia officinalis

*Euphrasia officinalis*, also known as eyebright, is native to temperate regions including parts of Europe, the Himalayas, and the Middle East. Traditionally, it has been widely used for treating eye-related conditions such as conjunctivitis, cataracts, ocular fatigue, and allergies, as well as upper respiratory issues like hay fever and sinusitis. Clinical studies support its safety and efficacy when used as an eye lotion for catarrhal and inflammatory conjunctival diseases. Beyond its ophthalmic uses, eyebright has demonstrated anti-inflammatory, antioxidant, antimicrobial, astringent, and hypoglycemic activities. The metabolic profiling reveals that the aerial parts are abundant in iridoid glycosides, such as aucubin, catalpol, geniposidic acid, and euphroside, which are considered key contributors to its anti-inflammatory bioactivity. The plant also contains phenylpropanoid glycosides like acteoside, various flavonoid glycosides (e.g., apigenin, luteolin, kaempferol), and phenolic acids, including caffeic and chlorogenic acid. These compounds likely act synergistically to support eyebright’s broad therapeutic effects, especially its inflammation-modulating and antioxidant properties [80,81].

The plant has demonstrated significant pharmacological potential through its anti-inflammatory, antioxidant, cytoprotective, and neuroprotective activities. *E. officinalis* ethanol leaf extracts were used to synthesize gold nanoparticles (EO-AuNPs), which markedly inhibited LPS-induced inflammation in RAW 264.7 macrophages by downregulating nitric oxide (NO), inducible nitric oxide synthase (iNOS), TNF-α, IL-1β, and IL-6 levels. (DOI: 10.2147/IJN.S199781). Ethanol extracts of E. officinalis have also demonstrated photoprotection in UVB-irradiated normal human dermal fibroblasts (NHDFs), where they scavenged reactive oxygen species (ROS), decreased MMP-1 and MMP-3 expression, preserved type I procollagen, and reduced apoptosis. These effects were mediated by downregulating MAPK and AP-1 signaling pathways, suggesting potential for anti-photoaging interventions [82]. Corneal cell models further confirmed Euphrasia’s ocular anti-inflammatory role. Ethanol and ethyl acetate extracts reduced IL-1β, IL-6, and TNF-α expression, while modulating nitric oxide levels and cytoskeletal integrity, supporting their suitability in ocular formulations [83]. Additionally, a commercial eye drop containing E. officinalis and Matricaria chamomilla protected human corneal epithelial cells from UVB-induced damage. The formulation reduced ROS and lipid/protein oxidative damage, promoted wound healing, and downregulated inflammatory markers (COX-2, IL-1β, iNOS), highlighting its potential in managing UV-induced ocular inflammation [84].

*E. officinalis* also showed systemic benefits. Aqueous leaf extract exhibited antihyperglycemic effects in alloxan-induced diabetic rats without affecting normoglycemic animals, suggesting glucose-lowering potential in an “on-demand” manner, specific to pathological states [85]. Neuroprotective studies revealed that multiple solvent extracts (notably, ethyl acetate and butanolic) enhanced neuronal viability under amyloid-beta stress, with strong antioxidant and anticholinesterase activity—suggesting applications in neurodegenerative disease management [86]. Finally, essential oil from *Euphrasia officinalis* demonstrated selective antimicrobial activity relevant to eye infections against Gram-positive bacteria (Staphylococcus aureus, S. epidermidis, Enterococcus faecalis) and Candida albicans [87].

## 3. Iridoids as Modulators of Inflammation: Evidence and Mechanistic Insights

The mechanistic features governing the anti-inflammatory properties of iridoid compounds have not been fully elucidated; however, various in vitro and in vivo studies provided solid evidence of their modulatory activity on multiple molecular targets in the inflammatory signaling network. Among the Bulgarian species examined in this review, most iridoid derivatives appear to interact with the complex cross-talk among key pro-inflammatory pathways, including antioxidant, apoptotic, NF-κB, JAK/STAT, and AMPK, and play a role in regulating immune responses, apoptotic mechanisms, and the antioxidant defense system (Figure 2) [88,89]. Furthermore, structure–activity relationship analyses suggest that specific structural features—such as electron-withdrawing carbonyl groups at C-11 and hydroxyl groups at C-10—enhance these bioactivities by promoting interactions with molecular targets involved in the inflammatory cascade. Structural modifications such as the introduction of a cinnamoyl group at C-6 of catalpol markedly amplify its anti-inflammatory potential. Moreover, synergistic interactions with other bioactives like flavonoids and anthocyanins further potentiate their antioxidant and anti-inflammatory effects, as demonstrated in models of hepatic inflammation [90].

### 3.1. Antioxidant Properties

Inflammation and oxidative stress are closely intertwined biological processes that play critical roles in the body’s response to harmful stimuli. While inflammation represents a protective immune response aimed at eliminating injurious agents and initiating tissue repair, its persistence can lead to tissue damage and pathological conditions. Oxidative stress occurs when the production of reactive oxygen species (ROS) surpasses the capacity of endogenous antioxidant defenses, resulting in molecular damage to lipids, proteins, and nucleic acids. Excess ROS not only contribute to cellular injury but also amplify inflammatory signaling, establishing a self-perpetuating cycle between oxidative stress and inflammation. This vicious cycle is a key driver in the pathogenesis of numerous chronic diseases, including metabolic, neurodegenerative, cardiovascular, and autoimmune inflammatory disorders [91,92].

Many iridoids have shown promise in countering both oxidative stress and inflammation.

In an in vitro model of oxidative stress, catalpol demonstrated significant antioxidant activity in human umbilical vein endothelial cells (HUVECs) by reducing intracellular reactive oxygen species (ROS) levels induced by hydrogen peroxide (H_2_O_2_) exposure [93]. In vivo, the neuroprotective potential of catalpol was evaluated in a cerebral ischemia model in gerbils, with treatment administered intraperitoneally immediately post-reperfusion. Compared to controls, catalpol significantly preserved cognitive function, reduced apoptosis, and protected hippocampal CA1 neurons, as evidenced by histological analysis. It enhanced endogenous antioxidant defenses by increasing glutathione peroxidase activity, lowering lipid peroxidation, and reducing nitric oxide synthase (iNOS) activity in both cortex and hippocampus. These findings suggest that the anti-inflammatory and neuroprotective effects of catalpol’s scaffold are at least partially owed to modulating oxidative and nitrosative stress pathways [94]. 

The broad-spectrum activity of catalpol has also been explored in the context of metabolic dysfunction in a rat model of diabetes induced by streptozotocin (STZ) and a high-fat, high-sugar diet. Intravenous iridoid treatment for 14 days improved body weight, normalized food and water intake, and favorably modulated plasma lipid profiles by reducing total cholesterol and triglycerides while increasing HDL cholesterol. Indeed, the authors reported enhanced activity of antioxidant enzymes, including superoxide dismutase (SOD), catalase (CAT), and glutathione peroxidase (GSH-Px), and a prominent reduction in malondialdehyde (MDA) levels as a major protective mechanism against morphological damage to pancreatic tissue [95]. 

The similar in structure iridoid glycoside aucubin is as extensively studied for its potential to restore redox homeostasis. In various experimental settings, it directly reduces levels of ROSs, malondialdehyde (MDA), and 4-hydroxynonenal (4-HNE), and effectively scavenges diverse free radicals, including superoxide, hydroxyl, and DPPH radicals, while boosting endogenous antioxidant defenses by key detoxifying enzymes (SOD, CAT, GSH-Px). Catalpol’s protective properties have been linked to these mechanisms across a variety of oxidative stress-related pathologies, including gastric mucosal injury, endothelial dysfunction, cardiac remodeling, diabetic neuropathy, and reproductive dysfunctions [96].

In triptolide-induced testicular injury in mice, aucubin pre-administration significantly preserved testicular weight, sperm morphology, and blood–testis barrier (BTB) integrity while restoring the balance of oxidative stress markers and endogenous antioxidants through activation of the Nrf2 signaling, promoting Nrf2 nuclear translocation and upregulation of downstream antioxidant enzymes. It also maintained BTB function by upregulating tight junction proteins such as ZO-1, Occludin, Claudin-11, and the gap junction protein Cx43 [97]. By mitigating oxidative stress and restoring redox homeostasis, aucubin significantly alleviated liver injury associated with NAFLD in a tyloxapol-induced mouse model of non-alcoholic fatty liver disease (NAFLD) and apoC-III-stimulated 3T3-L1 cells. Treatment with aucubin reduced oxidative stress by enhancing the activity of endogenous antioxidant enzymes (SOD) and lowering levels of oxidative markers like myeloperoxidase (MPO). Both in vivo and in vitro, aucubin activated key antioxidant and metabolic regulators, including Nrf2 and heme oxygenase-1 (HO-1) [98]. 

Asperuloside, a dominant iridoid in *Galium* and *Plantago* spp., has also displayed strong antioxidant activities. In inflamed colon tissue and macrophages, it restored redox balance by activating the Nrf2/HO-1 signaling pathway, increasing the expression of antioxidant enzymes such as HO-1 and NQO-1, and promoting Nrf2 nuclear translocation, supported by in silico data on its Nrf2 binding affinity [99]. Consistent with these findings, another study demonstrated that asperuloside significantly improved endothelium-dependent relaxations in both obese mice and IL-1β-treated aortas. It attenuated endothelial activation and reduced oxidative stress by scavenging mitochondrial reactive oxygen species (ROS). These vascular protective effects were primarily mediated through the upregulation of heme oxygenase-1 (HO-1) via activation of the Nrf2/HO-1 signaling pathway, independent of its anti-inflammatory actions. Notably, silencing HO-1 or endothelial-specific Nrf2 abolished these benefits, indicating the critical role of this pathway in mediating asperuloside’s vasoprotective effects [100].

The iridoid glycoside loganin, found in *G. verum*, has also displayed notable antioxidant activity in a rat model of cerebral ischemia–reperfusion (I/R) injury induced by middle cerebral artery occlusion. Immunological assays revealed that loganin treatment for 7 days mitigated oxidative damage by markedly reducing both ROSs and oxidative stress markers (MDA), while restoring the activity of superoxide dismutase (SOD) [101]. In a model of Aβ-induced neuroinflammation using BV-2 microglial cells, loganin demonstrated antiapoptotic and anti-inflammatory properties by targeting key upstream signaling pathways. It effectively inhibited microglial activation and suppressed the overexpression of Toll-like receptor 4 (TLR4), MyD88, and TRAF6 components that play a central role in apoptosis-related inflammatory signaling. By downregulating pro-inflammatory mediators such as TNF-α, IL-6, MCP-1, nitric oxide (NO), and prostaglandin E2 (PGE_2_), loganin reduced cellular stress that can lead to apoptotic cell death. Additionally, its modulation of iNOS and COX-2 expression suggests a role in balancing both oxidative and inflammatory responses, ultimately protecting microglial cells from Aβ-induced apoptotic damage [102].

The potent antioxidant activity of monotropein (*Galium* spp.) was found to be primarily mediated by direct scavenging of reactive oxygen species (ROS), restoring redox balance in oxidative stress-induced models. In osteoblasts exposed to H_2_O_2_, monotropein pretreatment enhanced mitochondrial membrane potential, reduced intracellular ROS levels, and restored sirtuin-1 expression. It also reversed H_2_O_2_-induced activation of apoptotic markers (caspase-3 and caspase-9) and suppressed NF-κB-mediated inflammatory signaling, thereby improving osteoblast viability [103]. Additionally, in a sepsis-induced cardiac injury model, monotropein significantly enhanced antioxidant defenses by increasing reduced glutathione levels (GSH), catalase (CAT) activity, and total antioxidant capacity, while reducing malondialdehyde (MDA) levels. This reduction in oxidative stress was accompanied by a decreased expression of inflammatory cytokines (TNF-α, IL-1β, IL-6) and improved myocardial cell survival [104]. Similarly, in diabetic retinopathy (DR) rats, monotropein treatment enhanced antioxidant levels, decreased angiogenic proteins, and reduced oxidative stress markers, suggesting its broad antioxidant potential [105].

Strong evidence also supports the antioxidant activity of the iridoids lamalbid and shanzhiside methyl ester, found in *Lamium album*, which induced a nearly 2-fold reduction in ROS production in MLP-treated cells, compared to unprotected control groups [106]. Recently, it was found that shanzhiside methyl ester and 8-O-acetyl shanzhiside methyl ester (8-OaS) displayed concentration-dependent inhibitory activity on the release of pro-inflammatory mediators, e.g., myeloperoxidase, elastase, and matrix metalloproteinase-9 enzymes, IL-8, and TNF-α, both in vitro and in vivo. Additionally, in f-MLP and LPS-stimulated rat neutrophils, both iridoid derivatives limited the production of LTB4, a potent chemoattractant and recruiter of polymorphonuclear leukocytes [107]. While primarily known for their sedating and sleep-promoting properties, verbenalin and hastatoside (*Verbena officinalis*) have also been shown to possess antioxidant and anti-inflammatory activities, contributing to the therapeutic potential of the species [108].

Similarly, *Lamium album*’s iridoid lamalbid (lamiridoside) demonstrated anti-inflammatory effects through complex mechanisms, including inhibiting the production of reactive oxygen species (ROS), scavenging free radicals, and inhibiting the release of inflammatory cytokines such as IL-8. Lamiridoside treatment has also been reported to reduce TNF-α secretion in LPS-induced cells by nearly 40% at low micromolar concentrations [106,109].

### 3.2. NF-κB Inhibition

The NF-κB pathway plays a central role in inflammation by regulating the expression of genes involved in the immune response, cell survival, and the recruitment of inflammatory cells. Functioning as a molecular switch, it is triggered by diverse stimuli to initiate a cascade of events leading to the production of inflammatory mediators, cytokines, chemokines, adhesion molecules, etc. This pathway is crucial for both innate and adaptive immunity and is implicated in a wide range of inflammatory, autoimmune, and malignant diseases [110]. 

In its canonical form, NF-κB is sequestered in the cytosol by inhibitory proteins (IκBs) and becomes activated upon their degradation via IκB kinase (IKK). Acting as a central downstream effector of the cellular stress and immune responses, NF-κB activation is triggered by a broad range of stimuli—including bacterial and viral pathogens (via pattern recognition receptors like Toll-like receptors), pro-inflammatory cytokines, and antigen receptor signaling—all of which converge on pathways that ultimately activate the NF-κB cascade. Disassociation from IκB enables the cytosolic NF-κB subunit to translocate into the nucleus and facilitate the trans-activation of multiple pro-inflammatory factors, including TNF-α, IL-1β, IL-6, COX-2, and iNOS [111].

Many iridoid glycosides have been recognized as potent modulators of this signaling pathway, producing anti-inflammatory, antioxidant, and cytoprotective effects, mostly by hindering the process of NF-κB nuclear translocation. 

Aucubin’s notable anti-inflammatory activity is largely owed to interfering with NF-κB signaling via inhibition of IκB degradation, consequently reducing COX-2 and TNF-α expression. Interestingly, the hydrolyzed form of aucubin generated through β-glucosidase treatment produced superior NF-κB suppression than the iridoid glycoside, possibly due to better cellular permeability and bioavailability, indicating that the aglycone moiety is the carrier of aucubin’s anti-inflammatory properties. Despite its low oral bioavailability, in vivo studies indicate that it is widely distributed in organs and exhibits pleiotropic organoprotective effects and an excellent safety profile [96,112,113]. Network pharmacology and gene expression analyses identified 116 target genes associated with aucubin’s action, many of which are involved in inflammation and oxidative stress regulation. Experimental validation in LPS-induced HepG2 cells and mouse models confirmed that aucubin significantly reduced pro-inflammatory cytokines such as TNF-α and IL-6, along with iNOS activity, by suppressing STAT3 phosphorylation and preventing NF-κB (p65) nuclear translocation [114]. Aucubin also exerted cardioprotective effects in rats against myocardial ischemia–reperfusion injury (MI/RI) by modulating both inflammatory and apoptotic pathways, reducing myocardial tissue damage and infarct size following ischemia–reperfusion events. The primary mechanism involved in cardioprotection was modulation of the STAT3/NF-κB/HMGB1 signaling cascade by altering phosphorylation at STAT3’s Ser727 and Tyr705 residues. As a result, the nuclear translocation of NF-κB p65 and the release of HMGB1 were inhibited, along with the downstream inflammatory cascade of pro-inflammatory cytokine production and tissue damage [115].

Catalpol is also notable in suppressing NF-κB activation and reducing levels of inflammatory stimuli, acting both upstream and downstream of the transcriptional factor (TNF-α, IL-1β, ROSs, NO, iNOS) [14,116]. Additionally, catalpol inhibits NF-κB and NLRP3 inflammasome signaling through AMPK/SIRT1 activation, contributing to the reported antifibrotic and nephroprotective effects [117]. There is also evidence of the nephroprotective effects of catalpol mediated by AMPK/SIRT1-dependent inhibition of NF-κB and the NLRP3 inflammasome, lowering IL-1β and TNF-α, preventing fibrosis, and preserving kidney function. Catalpol also benefits respiratory health by inhibiting mast cell degranulation and inflammatory enzymes (COX-2, LOX-1), reducing cytokines like IL-13 and eotaxin, and decreasing airway remodeling factors such as TGF-β1. In autoimmune chronic inflammatory diseases such as Sjögren’s syndrome, catalpol reduced lymphocytic infiltration and helped restore normal tissue function [117].

Similarly, verproside (catalpol derivative iridoid glycoside, found in *Veronica officinalis* and *Verbascum phlomoides*) has demonstrated strong anti-inflammatory activity in COPD pathology, closely linked to chronic inflammation and mucus hypersecretion. In an in vitro model, verproside inhibited the NF-κB transcriptional activity and the phosphorylation of upstream effectors such as IKKβ, IκBα, and TAK1, markedly suppressing NF-κB-driven MUC5AC expression, a central factor in mucus overproduction during airway inflammation. Similar results were obtained in in vivo settings, where verproside’s inhibition of this signaling cascade has reduced the transcription of pro-inflammatory mediators, ultimately mitigating airway inflammation and other fundamental pathophysiological mechanisms associated with COPD [118]. Furthermore, verproside interfered with the formation of the TNF-α receptor 1 signaling complex, comprising TRADD, TRAF2, RIP1, and TAK1, which plays a triggering role in NF-κB activation. As confirmed by in silico docking studies, the iridoid is embedded within the interface of TRADD and TRAF2 subunits and aborts downstream inflammatory responses [119]. 

Building on these preclinical findings, a standardized extract YPL-001 comprising veproside and five other iridoids as principal active components (piscroside C, isovanillyl catalpol, 6-O-veratroyl catalpol, catalposide, and picroside II) has progressed to Phase 2a clinical trials as a natural therapeutic candidate for chronic obstructive pulmonary disease, demonstrating translational potential from mechanistic insight to patient-oriented research. Comparative in vitro and in vivo screenings of the six iridoids present in YPL-001 revealed that verproside most potently suppressed TNF/NF-κB-driven MUC5AC expression and PMA/PKCδ/EGR-1-mediated IL-6 and IL-8 production in human airway epithelial cells. In a COPD mouse model, verproside significantly reduced lung inflammation and mucus overproduction via selective inhibition of PKCδ activation [118]. 

Asperuloside, a dominant iridoid in *G. verum*, was also shown to inhibit the activation of the NF-κB and MAPK signaling pathways, both in vitro and in various animal models of inflammation [120,121]. In LPS-induced RAW 264.7 cells, asperuloside suppressed the phosphorylation of IκB-α, thereby preventing NF-κB translocation to the nucleus and reducing the transcription of inflammatory genes. Asperulosidic acid exhibited a similar inhibitory pattern, suggesting that both compounds act on upstream signaling events to block NF-κB activation. This targeted suppression of NF-κB signaling contributes directly to their anti-inflammatory effects, including reduced production of cytokines such as TNF-α, IL-1β, and IL-6 [122]. Prominent NF-κB inhibition was also reported in an LPS-induced lung injury model, where asperuloside reduced myeloperoxidase activity and pro-inflammatory cytokines production, alleviating pulmonary edema and histopathological alterations in LPS-induced acute lung injury [120]. Furthermore, pretreatment with asperuloside greatly decreased the phosphorylation of extracellular signal-related kinases 1 and 2 (ERK1/2), c-Jun, N-terminal kinase (JNK), and p38 mitogen-activated protein kinase (p38MAPK), indicating that its anti-inflammatory effects involve concurrent suppression of both NF-κB and MAPK signaling pathways—two major and closely related regulators of inflammatory responses [120]. It has also been investigated for its therapeutic potential in inflammatory bowel diseases (IBD), particularly in chronic colitis. By disrupting NF-κB-mediated gene expression, asperuloside alleviated weight loss, reduced disease activity index (DAI), and improved colonic histopathology in a dextran sulfate sodium (DSS)-induced mouse model [99].

The NF-κB/HMGB1 inflammatory pathway was also found to be a molecular target for iridoids, more characteristic of the *Lamium album* species. Both Kang et al. and Zhang et al. reported that 8-O-acetyl shanzhiside methylester (8-OaS) protected against cerebral ischemia/reperfusion injury in diabetic rats by blocking TNF-α-induced NF-κB activation. In both TNF-α-stimulated neuronal cells and an in vivo diabetic stroke model, 8-OaS inhibited NF-κB and reduced HMGB-1 expression, leading to decreased brain damage and blood–brain barrier disruption [123]. Furthermore, 8-OaS was shown to exert antipyretic and anti-inflammatory effects via immune signaling and gut microbiota–neurotransmitter interactions. In a pyretic rat model, 8-OaS reduced the secretion of pyrogenic cytokines (IL-6, IL-1β) and central thermoregulatory mediator PGE2 through modulation of the TLR4/NF-κB and HSP70/NF-κB pathways. It also altered gut microbiota composition, notably decreasing *Alistipes* and *Odoribacter*, and showed correlations with neurotransmitter-related metabolites such as 5-hydroxytryptamine and tryptamine [124]. Further, 8-OaS also produced strong analgesic effects when administered intrathecally in a spinal nerve ligation (SNL) model of neuropathic pain, showing comparable efficacy to lidocaine and ketamine. Its therapeutic effects appear to be attributed to anti-inflammatory mechanisms in the spinal dorsal horn, notably through the inhibition of astrocyte activation, ERK phosphorylation, and TNF-α expression, which may further disrupt the downstream TNF-α/NF-κB pro-inflammatory signaling axis. These findings suggest that intrathecal 8-OaS administration is a promising approach for neuropathic pain treatment by targeting central inflammatory pathways [125]. The effects of 8-OaS on TNF-α-induced NF-κB signaling were also studied in SH-SY5Y cell cultures and in an in vivo ischemic diabetic stroke model. In vitro, 8-OaS inhibited TNF-α-induced NF-κB activation and reduced HMGB-1 expression. In the in vivo model, 8-OaS significantly mitigated brain injury and reduced histopathological damage and brain edema even after delayed administration (4 h post-ischemia). It also helped preserve the integrity of the blood–brain barrier in diabetic ischemic brains, contributing to the neuroprotective effects in acute ischemic stroke [123].

Inflammation is also a key characteristic of depression. Another *Lamium album*’s iridoid, shanzhiside methylester, exhibited strong anti-inflammatory activity both in a mouse model of depression induced by chronic stress and in LPS-ATP-induced inflammation in BV2 cells. Shanzhiside methylester treatment reduced inflammation-related markers such as TNF-α, IL-1β, IL-6, and Iba1, while also alleviating depression-like behaviors. These effects were linked to the suppression of the miRNA-155-5p/SOCS1/JAK2/STAT3 signaling axis, which is also tied to NF-κB signaling. MiRNA-155 is a microRNA nucleotide that has been found to promote inflammation by reducing SOCS1 levels, a negative regulator of cytokine signaling and a potent inducer of both JAK2/STAT3 and NF-κB pathways [126].

Harpagoside, one of the key iridoids found in *S. nodosa*, has demonstrated the capacity to inhibit lipopolysaccharide (LPS)-induced mRNA levels and expression of COX-2, as well as inducible iNOS in HepG2 cells. These inhibitory effects correlate with the upstream suppression of NF-kB activation, as pretreatment of cells with this phytochemical blocked the degradation of the inhibitory subunit IkB-α and the nuclear translocation of NF-kB. Furthermore, harpagoside dose-dependently inhibited LPS-induced NF-κB activity in a macrophage population, confirming its ability to block pro-inflammatory gene trans-activation [127,128]. Clinical evidence also supports the anti-inflammatory and analgesic properties of the iridoid. In a randomized, double-blind trial of 197 patients with chronic back pain, two daily doses of harpagoside-standardized *Harpagophytum* extract WS 1531 (600 mg and 1200 mg, containing 50 mg and 100 mg harpagoside, respectively) were compared with placebo over 4 weeks. The primary outcome was the number of patients who were pain-free for 5 days without using rescue medication (tramadol) during the final week. Among 183 completers, 3, 6, and 10 patients were pain-free in the placebo, 600 mg, and 1200 mg groups, respectively. Benefits were most notable in patients with shorter-duration pain or more severe, radiating pain, though some analyses suggested moderate doses (600 mg) also helped patients with less severe pain. The extract was well tolerated, with only occasional mild gastrointestinal symptoms reported [129].

Likewise, in models using macrophages and platelet systems, aucuboside has also demonstrated significant inhibition of leukotriene C4 (LTC4) release, indicating its potential to mitigate leukotriene-mediated inflammatory responses in allergic, atopic, and autoimmune diseases [50,130]. Other iridoids structurally related to those in *S. nodosa* have demonstrated selective inhibition of thromboxane B2 (TXB2) synthesis in human platelets. This mechanism is relevant for reducing thromboxane-mediated vasoconstriction and platelet aggregation, thereby playing a role in managing inflammatory processes occurring at the vascular level, in atherosclerosis, and other cardiovascular diseases. Structure–activity relationship analyses suggest that specific molecular substitutions, such as C-8 acetylation in 8-O-acetylharpagide, enhance anti-inflammatory potency through selective enzyme inhibition, particularly of thromboxane synthase [50].

Loganin (*Galium verum*) demonstrated a protective role against burn-induced intestinal injury by targeting inflammation through inhibition of the TLR4/NF-κB signaling pathway. In severely burned rats, loganin treatment significantly reduced intestinal levels of pro-inflammatory cytokines, including TNF-α, IL-6, and IL-1β, along with histological damage. These effects were linked to reduced expression of LPS-responsive TLR4 receptors and NF-κB inhibition, interfering with downstream inflammatory cascades involved in gut barrier dysfunction [131]. Loganin also shows promise as a therapeutic agent for cardiac hypertrophy and heart failure. A recent study demonstrated that this iridoid compound effectively inhibits Ang II-induced cardiac hypertrophy and damage in both H9C2 cells and mice. Its cardioprotective effects were attributed to the reduction in cardiac fibrosis, suppression of pro-inflammatory cytokines, and inhibition of key signaling proteins, including JAK2, STAT3, p65, and IκBα phosphorylation. Importantly, loganin exhibited no significant toxicity or adverse effects on normal cells and organs [132].

Another iridoid found in *Galium* spp., scandoside, exhibited potent anti-inflammatory effects primarily through inhibition of the NF-κB signaling pathway. In LPS-stimulated RAW 264.7 macrophages, scandoside significantly reduced the production of pro-inflammatory mediators, including nitric oxide (NO), prostaglandin E_2_ (PGE_2_), TNF-α, and IL-6, as well as the expression of pro-inflammatory enzymes iNOS and COX-2 at both protein and mRNA levels [133]. Similarly, monotropein, found in *Galium* spp., has demonstrated strong anti-inflammatory and cartilage-protective properties by inhibiting NF-κB signaling. In osteoarthritis models, monotropein reduced pro-inflammatory cytokines and chondrocyte apoptosis while decreasing matrix degradation. Specifically, it blocked IL-1β-induced NF-κB activation, resulting in the downregulation of MMP-3 and MMP-13 and increased COL2A1 expression, aiding in preserving cartilage integrity [134]. A mouse model of surgically induced osteoarthritis confirmed that monotropein inhibited cartilage matrix degradation, apoptosis, and pyroptosis by disrupting NF-κB signaling, leading to improved joint morphology and reduced OARSI scores [135]. In bone-resorption models, monotropein suppressed osteoclastogenesis through inhibition of NF-κB signaling and related upstream mediators. Treatment with monotropein reduced TRAP-positive osteoclast formation and downregulated RANK, RANKL, TRAF6, p65 phosphorylation, and IκBα degradation. These effects collectively attenuated inflammatory bone loss and inhibited the differentiation and activity of osteoclasts, highlighting monotropein as a potential therapeutic for osteoclastic bone diseases [136]. In atherosclerosis models, monotropein inhibited the proliferation and migration of vascular smooth muscle cells by suppressing NF-κB and AP-1 activation. This reduction in inflammatory signaling and oxidative stress significantly decreased atherosclerotic plaque formation and necrotic core size in LDLR–/– mice on a high-fat diet, suggesting a therapeutic potential of this iridoid in ischemic heart disease [137].

### 3.3. Modulation of the AMPK and Other Pathways Related to Metabolic Homeostasis

Energy homeostasis is tightly regulated by a complex network of hormonal, neural, and molecular signals that balance energy intake, storage, and expenditure. Disruptions in this balance, often driven by high-fat diets, sedentary lifestyles, or genetic factors, lead to excessive lipid accumulation, insulin resistance, and chronic low-grade inflammation—a hallmark of metabolic disorders such as obesity, type 2 diabetes, and non-alcoholic fatty liver disease (NAFLD). Adipose tissue plays a central role by secreting adipokines like leptin, resistin, and adiponectin, which not only regulate appetite and glucose metabolism but also influence inflammatory pathways such as NF-κB and JAK/STAT [138,139].

AMP-activated protein kinase (AMPK) emerges as a master regulator of energy homeostasis, linking metabolic regulation with anti-inflammatory and antioxidant responses. Activation of the AMPK signaling cascade promotes catabolic processes and inhibits anabolic pathways, particularly when cellular energy levels are low. Downstream signaling enhances glucose uptake, fatty acid oxidation, and mitochondrial biogenesis while reducing lipogenesis and inflammation, making it a critical therapeutic target in metabolic disorders such as obesity, diabetes, and non-alcoholic fatty liver disease (NAFLD) [140]. Key metabolic regulators, including adipokines like leptin, resistin, and adiponectin, interact with AMPK and related pathways to fine-tune energy expenditure, insulin sensitivity, and inflammatory responses. For instance, leptin, while improving glucose utilization and fatty acid metabolism, also acts as a pro-inflammatory mediator by activating NF-κB and JAK/STAT pathways; in contrast, adiponectin exerts strong anti-inflammatory and insulin-sensitizing effects and down-regulates pro-inflammatory cytokines by enhancing AMPK signaling [141,142,143]. Furthermore, adiponectin improves glucose utilization and fatty acid metabolism, partly via AMPK and downstream targets such as acetyl-CoA carboxylase (ACC) and glucose transporter 4 (GLUT4), while resistin is linked to insulin resistance and pro-inflammatory signaling [144]. Additionally, nuclear receptors such as PPARα and PPARγ, as well as oxidative stress regulators like Nrf2, play a synergistic role to restore metabolic balance, control lipid metabolism, reduce oxidative damage, and suppress inflammation [145]. Together, these pathways work to maintain energy balance and metabolic flexibility, providing several potential targets for treating metabolic syndrome and related disorders.

Many iridoid derivatives have shown promising effects on these metabolic and inflammatory pathways, tied to energy homeostasis, and effectively attenuate chronic low-grade inflammation commonly associated with obesity, type 2 diabetes, and metabolic syndrome. Furthermore, preclinical evidence suggests that iridoid glycosides, including geniposide, shanzhiside methylester, 8-OaS, morroniside, and catalpol, may act as small molecule GLP-1R agonists, indicating their broad therapeutic potential for diabetes and associated conditions, including obesity, fatty liver, hypertension, and cardiovascular diseases [11].

In an in vivo model of non-alcoholic fatty liver disease (NAFLD), aucubin ameliorated tyloxapol-induced hyperlipidemia, oxidative stress, and inflammation by improving lipid profiles (TC, TG, LDL, VLDL) and activating Nrf2, PPARα/γ, HO-1, and AMPK pathways [98]. In asthma models, it managed to reduce airway inflammation and histopathological damage by inhibiting the secretion of Th2-type cytokines and IgE and restoring cellular redox balance [146]. In another study, aucubin supported lipid metabolism and enhanced the antioxidant response through the activation of PPARα and PPARγ. These mechanisms collectively led to improved lipid profiles, as indicated by reductions in total cholesterol, triglycerides, LDL, and VLDL. Additionally, aucubin promoted phosphorylation of AMPKα/β and downstream targets such as acetyl-CoA carboxylase (ACC) and Akt, further contributing to redox balance and improved cellular metabolism [98].

The therapeutic potential of catalpol in inflammation related to metabolic syndrome has also been linked to AMPK signaling. In type 2 diabetes models induced by streptozotocin and/or a high-fat diet, catalpol activated AMPK in the liver, promoting glucose utilization, and improved mitochondrial function in skeletal muscle by upregulating PGC1. In diabetic mice, it further enhanced mitochondrial biogenesis in muscle via the AMPK–PGC1–TFAM pathway, boosting energy production. Furthermore, catalpol activates the insulin receptor substrate 1 (IRS-1)/PI3K/AKT/glucose transporter 4 (GLUT4) signaling pathway, enhancing insulin sensitivity in skeletal muscle and reducing fasting blood glucose levels [147,148]. It has also been demonstrated to modulate the intestinal microbiota and increase levels of glycolytic metabolites, which helped alleviate diabetes-related testicular damage. In an in vivo model of diabetic nephropathy, it slowed disease progression by inhibiting the RAGE/Ras homolog gene family member A (RhoA)/Rho-associated protein kinase (ROCK) pathway and suppressed angiogenesis in glomerular endothelial cells by targeting galectin-3 [149,150]. Another molecular target of catalpol is the insulin-like growth factor 1 (IGF-1)/IGF-1 receptor (IGF-1R) signaling pathway, which has been shown to downregulate mRNA expression of transforming growth factor-beta 1 (TGF-β1) and connective tissue growth factor (CTGF) in the renal cortex. In diabetic cardiomyopathy, catalpol reduced myocardial injury by the nuclear paraspeckle assembly transcript 1 (Neat1)/miR-140-5p/histone deacetylase 4 (HDAC4) pathway, and improved neointimal hyperplasia in hyperglycemic rats through the downregulation of monocyte chemotactic protein-1 (MCP-1) expression in the carotid artery [151]. It further promotes hypoglycemic effects by enhancing myogenic differentiation (MyoD)/myogenin (MyoG)-mediated myogenesis and upregulating protein kinase C gamma (PKCγ), and cytosolic carboxypeptidase 1 (CCP-1) in diabetic rat models. It also attenuated cognitive deficits and neuronal damage in diabetic rats by upregulating PKCγ and caveolin-1 (Cav-1) expression [148].

Asperuloside has also shown efficacy in mitigating obesity-related inflammation in multiple animal studies. In high-fat diet (HFD)-fed mice, asperuloside reduced circulating leptin levels and suppressed the hypothalamic expression of orexigenic peptides such as neuropeptide Y (NPY) and agouti-related peptide (AgRP), particularly under metabolic stress conditions. Leptin, an adipokine elevated in obesity, promotes low-grade systemic inflammation and contributes to insulin resistance through the activation of pro-inflammatory signaling pathways like JAK/STAT and NF-κB. Additionally, asperuloside lowered mRNA levels of pro-inflammatory cytokines (IL-1β, IL-6, TNF-α) in both the hypothalamus and liver and reduced plasma levels of plasminogen activator inhibitor-1 (PAI-1), another adipokine associated with inflammation, impaired fibrinolysis, and insulin resistance [152].

In another study, chronic administration of asperguloside in high-fat diet (HFD)-fed rats effectively attenuated inflammation and metabolic dysfunction associated with metabolic syndrome. Compared to unprotected control animals, asperguloside reduced body weight, visceral fat, food intake, and circulating levels of glucose, insulin, and lipids, while increasing adiponectin, a hormone with anti-inflammatory and insulin-sensitizing activities.

It further modulated multiple pathways involved in energy homeostasis. In white adipose tissue, asperguloside downregulated mRNA expression of isocitrate dehydrogenase 3α, NADH dehydrogenase flavoprotein 1 (Complex I), and fatty acid synthase, indicating reduced lipogenesis and alleviation of mitochondrial stress. In the liver, it upregulated carnitine palmitoyltransferase 1α and very-long-chain acyl-CoA dehydrogenase, promoting fatty acid oxidation. In skeletal muscle, iridoid treatment enhanced the expression of genes related to glucose uptake and mitochondrial function, including GLUT4, citrate synthase, and succinate dehydrogenase. Notably, asperguloside also elevated uncoupling protein 1 (UCP1) mRNA in brown adipose tissue, suggesting enhanced thermogenesis and energy expenditure [153].

An in silico study investigated the effects of asperuloside on inflammation and metabolic health by performing molecular docking analysis with adenosine receptors. Activation of specific adenosine receptor subtypes, such as A2A and A3, has been linked to reduced inflammation and enhanced thermogenesis in adipose tissue. Notably, asperuloside showed strong binding affinity to these receptors, as well as to other metabolic targets like TGR5 and AHR, contributing to its potential to suppress inflammation and promote energy expenditure [154].

Loganin has also shown promising anti-obesity effects by modulating metabolic function and inhibiting adipogenesis. In vitro, loganin suppressed lipid accumulation and adipocyte differentiation in both 3T3-L1 preadipocytes and adipose-derived stem cells by downregulating key adipogenic markers such as *Pparg*, *Cebpa*, *Plin2*, *Fasn*, and *Srebp1*. In vivo, oral loganin administration in ovariectomy (OVX)- and high-fat diet (HFD)-induced obese mice prevented excessive weight gain and reduced adipocyte enlargement and hepatic steatosis. Loganin also improved metabolic parameters by increasing serum leptin and insulin levels, suggesting enhanced endocrine and energy-regulating function [155]. Recent studies also indicate that loganin exerts pleiotropic hypoglycemic effects, improves insulin resistance, and protects pancreatic islet cells through complex mechanisms, demonstrated both in vivo and in vitro. Western blot analysis revealed that loganin upregulated the expression of key metabolic regulators, including adenosine 5′-monophosphate-activated protein kinase (AMPKα), sirtuin 1 (SIRT1), and PPAR-γ coactivator-1 alpha (PGC1α), in the skeletal muscle tissue of obese mice. Furthermore, 16S rDNA sequencing demonstrated that loganin reduced the diversity of intestinal microbiota and favorably altered the gut microbial composition in obese mice [156].

### 3.4. Antiapoptotic Activity

Apoptosis, or programmed cell death, plays a critical role in maintaining tissue homeostasis but can contribute to pathological damage when excessively activated, especially under oxidative stress or inflammatory conditions. Key mediators of apoptosis include caspases, the release of cytochrome c from mitochondria, and the balance between pro-apoptotic (e.g., Bax) and anti-apoptotic (e.g., Bcl-2) proteins, as well as cyclin-dependent kinases (CDKs) that regulate cell cycle progression and survival signals [157].

Iridoids have emerged as potent cytoprotective agents due to their ability to inhibit apoptotic pathways and enhance cell survival. The primary mechanisms by which they shift cells into a regenerative mode involve suppressing caspase activation, stabilizing mitochondrial membranes, upregulating Bax family proteins, and modulating pro-survival signaling pathways such as JAK2/STAT3 and PPARα.

Aucubin has exhibited significant anti-apoptotic activity in both in vitro (H9C2 cardiomyocyte cells) and in vivo (rat myocardial ischemia/reperfusion injury model) models, where it improved cardiac function by downregulating pro-apoptotic proteins such as caspase-3 and Bax, while upregulating the expression of the anti-apoptotic protein Bcl-2 [115]. Additionally, aucubin effectively suppressed triptolide-induced apoptosis by inhibiting the PERK/CHOP signaling pathway in a model of male reproductive dysfunction. Notably, siRNA-mediated silencing of Nrf2 abolished aucubin’s protective effects and failed to normalize CHOP overexpression, indicating that its antioxidative and antiapoptotic actions are intricately linked and largely dependent on Nrf2-mediated signaling [97].

Catalpol has also demonstrated prominent anti-apoptotic activity, particularly in an in vivo model of colitis. Modulation of the Sirtuin 1 (SIRT1) signaling axis was found to be the primary molecular target for the observed effects. SIRT1 is a critical regulator of cellular stress responses. SIRT1 is a NAD^+^-dependent deacetylase and a critical regulator of cellular stress responses that plays a key role in maintaining cellular homeostasis and regulating apoptosis, inflammation, and oxidative stress. In colitis, SIRT1 expression is typically downregulated, leading to increased endoplasmic reticulum (ER) stress and epithelial cell apoptosis. Catalpol restored impaired SIRT1 expression and redox homeostasis and decreased the expression of pro-apoptotic markers such as ATF6, CHOP, and caspase-12. It also reduces the acetylation of heat-shock factor-1 (HSF1), further promoting cellular resilience. Additionally, catalpol downregulated miR-132, a microRNA that directly targets SIRT1 for degradation, thus restoring SIRT1 levels and function [158].

Verbenalin also exhibited notable antiapoptotic effects by protecting against hepatic damage and mitochondrial dysfunction in alcohol-associated steatohepatitis (ASH). This effect was achieved by regulation of the MDMX/PPARα axis, linked to ferroptosis, a form of programmed cell death in response to oxidative stress. Studies in both alcohol-exposed mice and liver cells demonstrate that verbenalin directly targets the pro-survival MDMX protein, improves mitochondrial function, and inhibits ferroptosis-driven liver injury [159].

Cytoprotective and antiapoptotic properties have also been reported for loganin (*G. verum*) in the context of cerebral I/R injury. TUNEL staining showed that intragastrical administration of loganin effectively inhibited neuronal apoptosis by activating the pro-survival JAK2/STAT3 signaling pathway, as indicated by the increased phosphorylation of the JAK2 and STAT3 kinases [101]. Furthermore, loganin promoted Schwann cell survival under TNF-α-induced cytotoxic conditions by upregulating the cyclin D1–CDK4/6 complex and E2F-1-dependent survivin expression—key regulators of cell cycle progression and apoptosis resistance. By blocking the SMAD2 signaling pathway, loganin effectively reversed the inhibitory effects of TNF-α on Schwann cell survival and peripheral nerve repair. By enhancing survivin levels, a critical pro-survival protein, loganin helped maintain cell survival and viability and promoted regenerative healing [160]. Furthermore, loganin has demonstrated protective effects against diabetic kidney disease (DKD), largely through its regulation of inflammasome-mediated pyroptosis. In vivo, loganin reduced fasting blood glucose, serum creatinine, and urea nitrogen levels, while improving renal histopathology in DKD mice. Notably, it inhibited the activation of the NLRP3 inflammasome, a key driver of inflammation in DKD, thereby downregulating pyroptosis-related proteins such as Caspase-1 and Gasdermin D (GSDMD), and decreasing circulating IL-1β and IL-18 levels. In vitro, loganin suppressed high glucose- and polyphyllin VI-induced pyroptosis in HK-2 cells by reducing ROSs and blocking NLRP3 inflammasome activation [161].

Loganin also exerted strong antiapoptotic effects in an SH-SY5Y in vitro model of H_2_O_2_-induced oxidative stress and neuronal injury. Pretreatment with loganin significantly improved cell viability, reduced LDH release, lowered ROS production, and preserved mitochondrial membrane potential—key indicators of reduced apoptotic damage. Evidently, loganin inhibited H_2_O_2_-induced activation of apoptotic markers, including cleaved PARP and the executioner caspase-3 of programmed cell death, restored the ratio between antiapoptotic Bcl-2 and proapoptotic Bax proteins, and stabilized mitochondrial integrity by blocking the release of cytochrome c from mitochondria [162]. A similar pharmacodynamic profile was observed in a mouse model of sepsis-induced acute kidney injury (AKI), as well as LPS-stimulated HK2 cells, where loganin treatment elicited pronounced antiapoptotic effects [163].

*Galium* spp.’s iridoid monotropein has demonstrated comparable anti-apoptotic properties in various disease models. In IL-1β-stimulated chondrocytes, monotropein reduced apoptosis and pyroptosis while promoting cell proliferation. In a mouse model of osteoarthritis, monotropein treatment decreased cartilage cell death, likely due to NF-κB inactivation and subsequent suppression of apoptosis-inducing factors [135]. In sepsis models, monotropein reduced myocardial apoptosis by modulating pro- and anti-apoptotic proteins, decreasing Bax and cleaved caspase-3 levels, while increasing the expression of anti-apoptotic Bcl-2 members [104]. Similarly, monotropein treatment lowered pro-apoptotic protein levels in retinal tissue and improved histological outcomes in STZ-induced diabetic retinopathy rat models [105].

### 3.5. Immunomodulatory Activity

Numerous studies have demonstrated that iridoid compounds can effectively modulate both innate and adaptive immune responses, highlighting their potential in managing chronic and acute inflammatory conditions. By regulating key immune cell functions—such as macrophage polarization, cytokine production, and immune cell recruitment—iridoids help restore immune homeostasis and prevent excessive inflammatory damage.

Both in vivo and in vitro investigations demonstrated that the aucubin treatment remarkably inhibited the activation of the NF-κB-NOD-like receptor protein 3 (NLRP3) inflammasome pathway in chondrocytes, evidenced by decreased expression of phosphorylated P65 (p-P65), NLRP3, and Caspase-1. In a murine model of intervertebral disc degeneration (IDD), aucubin treatment attenuated cartilage endplate degeneration, suggesting its potential as a therapeutic agent for IDD [164]. These molecular pathways were similarly implicated in aucubin’s hepatoprotective effects in diabetic models. In high-fat diet/STZ-induced diabetic mice and high glucose/TGF-β1-stimulated LX-2 cells, aucubin treatment suppressed NLRP3 inflammasome activation by inhibiting IRE1α/TXNIP signaling, resulting in reduced activation of hepatic fibrogenic stellate cells and alleviation of inflammation and liver injury [165].

The therapeutic potential of catalpol has been studied in the context of allergic inflammation both in vitro and in vivo. In IgE/ovalbumin (OVA)-stimulated mouse bone marrow-derived mast cells, it significantly reduced mast cell degranulation without altering the expression of the high-affinity IgE receptor (FcεRI) or causing mast cell death. In a mouse model of IgE/OVA-induced asthma, oral administration of catalpol lowered IgE levels, airway hyperresponsiveness, and lung infiltration by eosinophils and neutrophils. Histological analysis showed reduced mast cell recruitment and increased mucus production in the lungs of treated mice. Furthermore, catalpol decreased levels of Th2 cytokines (IL-4, IL-5, IL-13), PGD2, eotaxin-1, and CXCL1 in bronchoalveolar lavage fluid, compared to the allergic control group [166]. In another murine model of bronchial asthma, catalpol-treated animals exhibited a significantly reduced total cell count in bronchoalveolar lavage fluid compared to the unprotected control group. Additionally, eosinophil and neutrophil counts were markedly decreased in response to catalpol treatment, alongside MDA levels in lung tissue homogenates. Cytokine analysis revealed significantly lower IL-4 and IL-5 concentrations in the catalpol group, while IFN-γ levels were elevated [167]. Both interleukins amplify allergic inflammation and are central to the pathogenesis of eosinophilic asthma, driving type 2 (Th2) immune responses. IL-4 primarily promotes Th2 cell differentiation and B-cell class switching to IgE, facilitating allergic sensitization and airway inflammation. IL-5, on the other hand, is crucial for the growth, recruitment, activation, and survival of eosinophils, which release cytotoxic proteins and pro-inflammatory mediators that contribute to airway hyperresponsiveness, mucus hypersecretion, and tissue damage. Due to their critical roles in sustaining type 2 inflammation, both cytokines have emerged as key therapeutic targets, with monoclonal antibodies designed to modulate their activity showing significant clinical efficacy in reducing asthma severity and exacerbations [168]. In light of these findings, catalpol may be considered a promising candidate for the clinical managment of asthma and other Th2-mediated inflammatory conditions.

Verbenalin is another perspective iridoid in coping with lung inflammatory conditions. In vitro, verbenalin suppressed LPS- and IgG IC-mediated activation of inflammatory signaling cascades by engaging with the GPR18 receptor, as indicated by molecular docking and molecular dynamics simulations. Additionally, it blocked the IgG IC-induced formation of neutrophil extracellular traps (NETs), thereby reducing excessive inflammatory responses and preventing subsequent tissue damage. These results collectively establish verbenalin as a potent “phytoresolvin” that promotes the resolution of inflammation in acute lung injury and sepsis [169]. Another study reported that verbenalin reduced cell injury and inflammation in human coronavirus 229E (HCoV-229E)-infected macrophages and improved lung inflammation in a mouse model of pneumonia. Both in vivo and in vitro, verbenalin effectively blocked the activation of the NLRP3 inflammasome and downregulated the expression of IL-1β, caspase-1, and gasdermin D (GSDMD). Furthermore, the use of a mitophagy inhibitor or RNA interference (RNAi) decreased the inhibitory capacity of verbenalin on NLRP3 inflammasome signaling, indicating that its anti-inflammatory activity is regulated at an upstream level, namely via the PINK1/Parkin mitophagy pathway. The latter serves as a crucial mitochondrial quality control mechanism that identifies and induces autophagy in damaged mitochondria. By targeting this pathway, verbenalin effectively reduced mitochondrial dysfunction and the subsequent activation of inflammatory signaling [170].

*Galium verum*’s iridoid constituents have also demonstrated multifaceted immunomodulatory activities in chronic inflammatory diseases, including IBD, atopic dermatitis AD, osteoarthritis, pancreatitis, neuroinflammation, and diabetes.

Numerous in vitro and in vivo studies have confirmed that loganin is quite effective in suppressing macrophages’ M1 polarization and their subsequent differentiation into a pro-inflammatory phenotype, characterized by the overproduction of cytokines such as TNF-α, IL-1β, and IL-6, as well as inflammatory mediators like iNOS and COX-2. Studies have shown that loganin not only suppresses M1 polarization but also promotes M2 polarization, providing a foundation for managing oxidative stress and immune-inflammatory disorders [101,171]. In a mouse model of DSS-induced ulcerative colitis, loganin reduced the number of F4/80+iNOS-positive M1 macrophages and downregulated pro-inflammatory mediators such as MCP-1, CXCL10, and COX-2 in colonic tissues. Notably, suppression of CXCL10, a key chemokine in M1 macrophage recruitment and activation, suggests that loganin disrupts the self-perpetuating loop of macrophage-driven inflammation. These effects contributed to improved disease outcomes, including reduced weight loss, colon shortening, and histopathological damage [172]. It was also found to alleviate the inflammatory symptoms associated with NASH by inhibiting NLRP3 inflammasome activation [173].

Hastatoside, a sleep-promoting iridoid glycoside from *Verbena officinalis*, has also been ascribed analgesic, anti-inflammatory, and tissue-protective effects. A recent study demonstrated its protective role against carbon tetrachloride (CCl_4_)-induced liver fibrosis in mice. The molecular target of the iridoid was reported to be the glycogen synthase kinase-3β (GSK-3β) enzyme. By enhancing its activity, hastatoside treatment led to a reduced expression of fibrosis-related proteins and inhibited the activation and proliferation of hepatic stellate cells [174].

Geniposidic acid (*Galium* spp.) has shown significant anti-inflammatory activity by modulating immune cell recruitment and cytokine expression, as demonstrated in an in vitro THP-1 monocyte/macrophage inflammatory model induced by PMA and LPS. Treatment with geniposidic acid markedly reduced the secretion of pro-inflammatory cytokines, including IL-1, IL-6, IL-8, and TNF-α. Proteomic profiling revealed that geniposidic acid regulates multiple biological processes and signaling pathways associated with immune cell activation and chemotaxis. Similar to other iridoids, it inhibited the expression of chemoattractants (CXCL10, an eosinophil, monocyte and T-cell recruiter), enzymes (PPM1H, a phosphatase involved in the regulation of autophagy) and receptors (TLR5, a pattern recognition receptor expressed in a variety of cell types), balancing the regulation of innate immunity and inflammation [175].

To provide a comprehensive overview, Table 1 summarizes the main iridoid constituents identified across the studied species, along with their key molecular mechanisms and associated biological effects, consolidating the evidence discussed above into a comparative framework.

## 4. Current Limitations and Future Perspectives for Iridoid-Based Therapeutics

Despite the compelling therapeutic scope of iridoid compounds, their clinical translation still encounters critical challenges, including a lack of solid evidence from clinical trials, inadequacy in safety assessment and toxicity risks, as well as insufficient pharmacokinetic data and limited bioavailability.

### 4.1. Safety Profile and Genotoxicity

While iridoids are often described as relatively safe, robust toxicological assessments remain limited.

Existing research evidence suggests that catalpol is well-tolerated and exhibits low toxicity in both animal models and human patients. Preclinical studies in rodents have demonstrated a favorable safety profile. Acute toxicity testing in mice showed no observable toxic effects or behavioral abnormalities at oral doses up to 1000 mg/kg, and the median lethal dose (LD_50_) via intraperitoneal injection was determined to be 206.5 mg/kg. Long-term intravenous administration in Wistar rats did not result in significant changes in biochemical markers or histopathological alterations in major organs, indicating minimal chronic toxicity [176]. According to the only available clinical evaluation of catalpol in colon cancer patients, intraperitoneal injections of 10 mg/kg catalpol twice daily for 12 weeks showed good tolerability. Only mild, non-fatal adverse effects, such as nausea, vomiting, and constipation, were reported [177].

Toxicity assessments of aucubin have similarly indicated a low-toxicity profile in both acute and long-term studies in rodents. In mice, intraperitoneal injection of aucubin at doses up to 900 mg/kg did not cause fatal events, although minor alterations in serum enzymes and triglyceride levels were noted at doses above 300 mg/kg. Repeated dosing (20–80 mg/kg) had no significant impact on liver function, metabolic parameters, or liver histology. In rats, a single intraperitoneal dose up to 100 mg/kg was non-lethal, though transient paralysis was observed at the highest dose. Intragastric administration of aucubin at an extremely high dose of 40 g/kg in mice led to only mild, reversible symptoms, such as reduced activity, soft stools, and fat feces, with no pathological changes in major organs. Moreover, a 6-month oral toxicity study in rats (200–800 mg/kg) revealed no adverse effects on body weight, behavior, hematological parameters, or organ structure, and no signs of chronic or delayed toxicity [96]. These findings collectively support aucubin’s classification as a compound with a high safety margin under both acute and chronic exposure conditions.

In contrast, concerns have been raised regarding the genotoxic potential of the iridoid verminoside (*Veronica officinalis*) and the phenylethanoid glycoside verbascoside (*Verbascum phlomoides*). A study employing FDA-recommended genotoxicity assays demonstrated that both compounds induced genetic damage in human lymphocytes [178]. While direct experimental data on the genotoxicity of aucubin are currently lacking, its chemical structure and the high reactivity of its aglycone raise concerns about potential genotoxic liability. It has been noted that aucubin’s aglycone may produce reactive dialdehydes under acidic conditions, which are capable of forming Schiff-base adducts with proteins and potentially DNA, indicating a plausible mechanism for DNA damage. Computational structure–activity relationship (QSAR) models further support this concern, showing mutagenic and DNA-binding alerts for aucubin and structurally related iridoid glycosides following aglycone formation [179]. Further risk-benefit assessments are necessary to establish acceptable exposure thresholds.

Notably, both pro-apoptotic and cytoprotective outcomes have been reported for certain iridoid-rich species, depending on context. For example, *Plantago major* leaf extracts (rich in aucubin) show selective cytotoxicity against certain breast cancer cells, yet aucubin alone was non-toxic except at very high concentrations and robustly suppressed neutrophil ROS production [180,181]. Meanwhile, verbascoside (*Verbascum* spp.) induced apoptosis in colorectal and breast cancer cells via p53 activation but remains safe for normal cells [182,183]. These findings build the emerging concept that iridoids exhibit biased effects on cell proliferation, depending on the experimental conditions (e.g., physiological or pathological settings), preserving healthy tissues through anti-inflammatory and anti-apoptotic pathways (e.g., NF-κB inhibition), while inducing apoptosis selectively in malignantly transformed cells.

Given the limited toxicological data available for many other iridoid derivatives, further research is necessary to comprehensively assess their safety profiles and support their potential therapeutic use.

### 4.2. Pharmacokinetics and Bioavailability

The oral bioavailability of iridoid glycosides remains a significant barrier to their clinical translation, largely due to poor absorption, instability in the gastrointestinal (GI) tract, extensive first-pass metabolism, and interindividual or pathological variability. Furthermore, the lack of pharmacokinetic data for many iridoid compounds remains a fundamental barrier to their therapeutic development.

For instance, verproside, a catalpol derivative, demonstrated extremely low oral bioavailability in rats (0.3–0.5%) and short systemic half-life, likely due to extensive hepatic biotransformation. On the contrary, the systemic clearance of verproside after intravenous administration of high doses (10 mg/kg) was significantly reduced, likely due to saturable hepatic metabolism, suggesting a shift toward zero-order kinetics at higher concentrations, which further limits its therapeutic window [184].

Catalpol is somewhat better characterized. When administered orally to rats at 50 mg/kg body weight, catalpol was rapidly absorbed (Tmax ~1.3 h) but was also eliminated quickly, with a plasma half-life of approximately 1.2 h. Distribution studies showed high concentrations in the kidney, followed by the liver, heart, and lung within minutes of administration. It has also been demonstrated that catalpol is capable of crossing the blood–brain barrier, albeit at low levels (CSF/plasma AUC ratio ≈ 5.8%) [185]. These pharmacokinetic characteristics, fast absorption and rapid elimination, constrain its therapeutic potential and necessitate frequent dosing or modified delivery strategies.

Aucubin has shown moderate oral bioavailability (~20%) but also faces pharmacokinetic challenges. Its low lipophilicity and instability in gastric pH contribute to poor absorption, and significant first-pass metabolism in the gastrointestinal tract and liver reduces systemic exposure. The elimination half-life of aucubin varies significantly depending on formulation, ranging from ~2 to 7 h. Following intravenous administration, aucubin was widely distributed throughout various organs, including the kidney (which showed the highest concentration), liver, heart, spleen, lungs, and brain, and was excreted primarily unchanged. Interestingly, after oral dosing, aucubin appeared to bind covalently to serum proteins such as albumin, forming a stable systemic pool of the compound [179]. Notably, context-dependent pharmacokinetics has been reported for some iridoid compounds. Under pathological conditions, such as chronic kidney disease (CKD), the absorption, bioavailability, and distribution of loganin appear to be significantly altered. Studies using UPLC–MS have shown that loganin’s Cmax, Tmax, and AUC all increase in CKD models compared to healthy controls, likely due to downregulation of intestinal P-glycoprotein (P-gp), which normally limits its absorption [186]. These findings suggest that disease status can significantly impact iridoid pharmacokinetics, and personalized dosing or monitoring may be necessary in clinical conditions.

In addition, interactions between iridoids and drug-metabolizing enzymes or transporters are poorly understood, and their impact on co-administered pharmaceuticals remains unexamined. In rat liver microsomes and hepatocytes, aucubin was not converted into its aglycone and did not inhibit 7-ethoxycoumarin O-deethylase activity, indicating that it is not metabolized by liver microsomes and does not influence hepatic cytochrome P450 activity [187].

Limited data also exists regarding catalpol’s modulatory activity on human liver CYP450 enzymes. A recent study evaluated catalpol’s inhibitory effects on eight major human liver CYP isoforms using human liver microsomes. The results showed that catalpol selectively inhibited CYP3A4, CYP2E1, and CYP2C9, with IC_50_ values of 14.27, 22.4, and 14.69 μM, respectively, while other isoforms were unaffected. Enzyme kinetics revealed that catalpol acts as a non-competitive and time-dependent inhibitor of CYP3A4 and a competitive inhibitor of the CYP2E1 and CYP2C9 isoenzymes. These in vitro findings suggest the potential for catalpol to cause pharmacokinetic drug interactions, particularly with co-administered CYP3A4, CYP2E1, and CYP2C9 drug substrates [188]. However, numerous dietary sources and natural compounds have also been reported to modulate CYP450 enzymes and P-glycoprotein (P-gp) activity, indicating that such interactions are not unique to iridoids and should be assessed within the broader context of dietary and phytochemical influences on drug metabolism. Further in vivo studies are warranted to assess the clinical relevance of these pharmacokinetic interactions.

### 4.3. Chemical Modifications and Technological Approaches to Improve Iridoid Therapeutic Potential

The outlined pharmacokinetic limitations of iridoids highlight the urgent need for the development of efficient delivery platforms for these phytocompounds to ensure effective therapeutic concentrations at their molecular targets. Consequently, formulation approaches to protect iridoids from degradation, enhance their permeability, and enable targeted or sustained delivery become a critical prerequisite for their clinical translation into viable treatments.

Nanoparticle-based platforms, such as lipid nanoparticles, present a powerful strategy to overcome the inherent physicochemical challenges of iridoid glycosides, including aucubin and catalpol. One study successfully developed water-in-oil-in-water (W/O/W) lipid nanoparticles via emulsification-sonication, optimizing parameters through factorial design to achieve stable carriers with mean particle sizes under 100 nm, low polydispersity, and robust stability. The innovative formulation achieved impressive encapsulation efficiencies: approximately 89% for aucubin and 77% for catalpol [189]. Further, these nanoparticles were integrated into hydrogel formulations, resulting in enhanced physicochemical stability and improved functional outcomes. In vivo skin studies demonstrated superior moisture retention, decreased transepidermal water loss, and enhanced regeneration of the stratum corneum’s protective barrier, underscoring the dual role of lipid nanoparticles in protecting and enhancing the delivery of iridoid molecules [190].

Other polymer-based strategies, such as biodegradable microspheres, could also serve effectively as controlled-release platforms for iridoid delivery. These systems, often formulated from PLA or PLGA, enable sustained drug release over days to weeks, reducing dosing frequency, improving compliance, and maintaining therapeutic drug levels—critical advantages for compounds with short half-lives or rapid elimination [191].

For the central nervous system, active iridoids like catalpol, brain-targeting delivery systems offer further advantages. Nanoparticles, whether lipid-based, polymeric, or liposomal, can be engineered with surface modifications (e.g., PEGylation, peptide ligands) to achieve extended circulation and facilitate blood–brain barrier (BBB) penetration. Such functionalization supports both passive and receptor-mediated transcytosis across the BBB, enabling more effective CNS delivery and potentially greater therapeutic specificity [192].

Finally, the design of lipophilic prodrugs or enzyme-activated analogs of promising iridoid glycosides or their aglycones can enhance oral absorption and metabolic stability. By masking hydrophilic groups and optimizing molecular lipophilicity, structural modifications could greatly improve permeability and reduce first-pass metabolism. Natural iridoids typically contain a hemiacetal structure with a reactive hydroxyl group at the C-1 position, often found as glycosides in plants. This hydroxyl group on the sugar moiety contributes to their hydrophilicity, limiting membrane permeability. To address this, structural modifications, such as acetylation or benzylation of hydroxyl groups, are commonly applied to increase lipophilicity and improve oral bioavailability. For iridoid aglycones, which often feature multiple hydroxyl groups at C-1, oxidation is used to enhance compound stability. However, the dihydrofuran ring, central to iridoid structure, can negatively affect stability. Oxidation of the C-1 hydroxyl may lead to the formation of a lactone with the O-2 group, which tends to be chemically unstable. A more effective strategy involves converting the lactone to a lactam via substitution at the O-2 position. This modification has been shown to substantially improve both chemical stability and biological activity, offering a promising route for the development of more robust iridoid-based therapeutics [193].

While medicinal-chemistry prodrug designs for iridoids remain underexplored, many iridoid glycosides act as natural prodrugs. Their biological activity often arises after enzymatic deglycosylation by host enzymes or gut microbiota, producing more lipophilic, cell-permeable aglycones or phenolic fragments that serve as the actual active principals.

## 5. Conclusions

This review presents a comprehensive and in-depth overview of the anti-inflammatory potential of iridoid-containing medicinal plants native to the Bulgarian flora. We have detailed the phytochemical profiles of ten selected species, highlighting their iridoid constituents and associated biological activities. Particular emphasis was placed on the in vitro and in vivo evidence demonstrating the anti-inflammatory, antioxidant, antiapoptotic, and disease-modifying effects of key iridoids such as catalpol, aucubin, verbenalin, and hastatoside. Furthermore, we explored the mechanistic pathways through which these compounds exert their pharmacological actions, including modulation of NF-κB, Nrf2, STAT3, and ferroptosis-related signaling. By integrating both compound-specific and plant-based findings, this review underscores the therapeutic relevance of iridoids and their potential role as natural modulators of chronic inflammation. Taken together, our synthesis supports the value of these species as promising sources for the development of complementary anti-inflammatory agents.

## Figures and Tables

**Figure 1 molecules-30-03456-f001:**
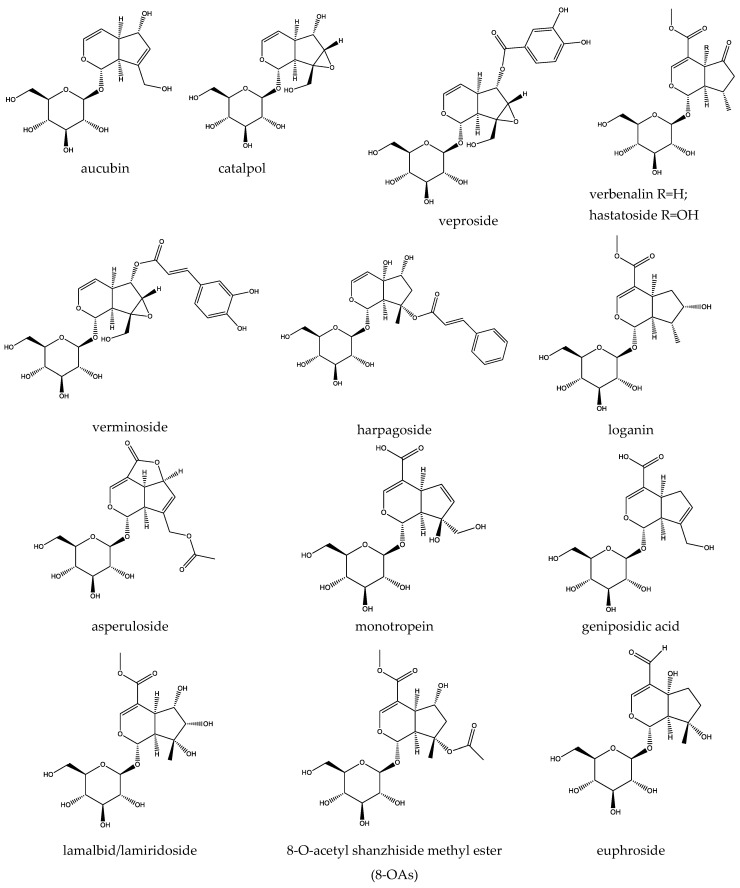
Main iridoid structures in the selected Bulgarian species.

**Figure 2 molecules-30-03456-f002:**
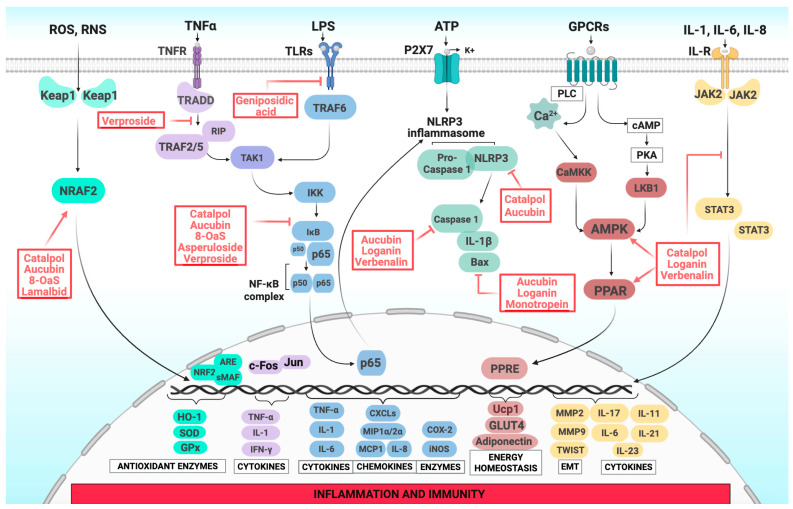
Molecular mechanisms of the anti-inflammatory activity of iridoid compounds, targeting NF-κB, AMPK, JAK/STAT, and antioxidant pathways. Created in BioRender. Mihaylova, R. (2025).

**Table 1 molecules-30-03456-t001:** Main iridoids in the above-cited species native to the Bulgarian flora and their molecular mechanisms and biological effects.

Iridoid	Plant Species	Molecular Mechanisms	Biological Effects	Ref.
Aucubin	*Veronica officinalis*, *Plantago* spp., *Verbena officinalis*, *Verbascum phlomoides*, *Euphrasia officinalis*	- scavenges ROS, MDA, 4-HNE - enhances SOD, CAT, GSH-Px - activates Nrf2/HO-1 pathway - inhibits NF-κB and STAT3 phosphorylation - downregulates TNF-α, IL-6, iNOS - modulates tight junction proteins (ZO-1, Occludin, Claudin-11, Cx43) - PPARα and PPARγ activation - NLRP3 inflammasome inhibition - suppressed caspase-1 activation - caspase-3 and Bax down-regulation	Antioxidant, cardioprotective, hepatoprotective, anti-allergic in airway inflammation, chondroprotective	[95,98,114,146,164]
Catalpol	*Veronica officinalis*, *Plantago* spp., *Scrophularia nodosa*, *Verbascum phlomoides*, *Euphrasia officinalis*	- reduces ROS and iNOS - increases SOD, CAT, GSH-Px - inhibits NF-κB and NLRP3 - reduces IL-1β, TNF-α, COX-2, LOX-1 - suppresses mast cell degranulation - GLP-1R activation - activates AMPK pathway	Antioxidant, anti-inflammatory, bronchoprotective, nephroprotective, hepatoprotective, alleviating effects in diabetes and its related multiorgan complications, neuroprotective	[11,94,117,118,148,158,166]
Verproside	*Veronica officinalis*, *Verbascum phlomoides*	- inhibition of NF-κB signaling - reduced MUC5AC expression - inhibited TNF-α signaling	Anti-inflammatory, pulmoprotective	[118,119]
Verbenalin	*Verbena officinalis*	- ROS detoxification - regulation of the MDMX/PPARα axis - caspase-1 inhibition - NLRP3 inflammasome suppression and “phytoresolving” properties - AMPK activation	Antioxidant, antiapoptotic, hepatoprotective, pulmoprotective, antiviral	[159,169,170]
Hastatoside	*Verbena officinalis*	- antioxidant activity - reduced expression of fibrosis-related proteins	Antioxidant, anti-inflammatory, reduces airway inflammation and mucus hypersecretion in COPD, neuroprotective, antimicrobial	[108,174]
Harpagoside	*Scrophularia nodosa*	- inhibits COX-2 and iNOS expression - prevents IκB-α degradation - blocks NF-κB nuclear translocation	Anti-inflammatory	[127,128]
Aucuboside	*Scrophularia nodosa*	- inhibits LTC_4_ release - reduces TXB_2_ synthesis - modulates leukotriene and thromboxane pathways	Anti-inflammatory, anti-colitis	[50,130]
Loganin	*Galium* spp.	- reduces ROS, MDA - enhances SOD and NRf2 activity - inhibits TLR4/MyD88/NF-κB - Downregulates TNF-α, IL-1β, IL-6, iNOS, COX-2 - inhibits NLRP3 inflammasome - modulates JAK/STAT and AMPK signaling	Antioxidant, anti-inflammatory, neuroprotective, cardioprotective, anti-obesity effects, nephroprotective, antipsoriatic, anti-colitis	[101,102,131,132,155,161,163,171,172,173]
Asperuloside	*Galium* spp. *Plantago* spp.	- activates Nrf2/HO-1 - suppresses NF-κB and MAPK (ERK, JNK, p38) - reduces TNF-α, IL-1β, IL-6 - inhibits IκBα phosphorylation - reduced leptin secretion -up-regulates GLUt-4 expression	Antioxidant, vasoprotective, anti-colitis, pulmoprotective, anti-obesity	[99,100,120,121,152]
Monotropein	*Galium* spp.	- scavenges ROS - enhances mitochondrial potential, SIRT1, GSH, CAT - suppresses NF-κB and AP-1 - inhibits caspase-3/9, RANKL/TRAF6 - reduced MMP activity - antiapoptotic action	Antioxidant, retinoprotective, ameliorates joint and cartilage inflammation, antiosteoporotic, anti-atherosclerotic	[103,105,134,135,136,137]
Scandoside	*Galium* spp.	- inhibits NF-κB - decreases NO, PGE_2_, TNF-α, IL-6, iNOS, COX-2 - reduced MAPK signaling	In vitro anti-inflammatory	[133]
Geniposidic acid	*Galium* spp.	- reduced expression of IL-1, IL-6, IL-8, and TNF-α; - chemoattractants (CXCL10) and Toll-like receptors (TLR5)	In vitro anti-inflammatory	[175]
Lamalbid/	*Lamium album*	- reduces ROS levels - inhibits IL-8 and TNF-α secretion	Antioxidant, anti-inflammatory	[106,109]
lamiridoside				
8-O-acetyl shanzhiside methyl ester (8-OaS)	*Lamium album*	- NF-κB pathway inhibition - reduced LTB4 secretion - modulates TLR4/NF-κB, HSP70/NF-κB pathways - modulates gut–brain axis - modulates JAK2/STAT3 axis in neuroinflammation and depression - GLP-1R agonists	Anti-inflammatory, neuroprotective, antidepressant	[123,126]

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
