# Peer review of "Targeting Inflammation with Natural Products: A Mechanistic Review of Iridoids from Bulgarian Medicinal Plants"

_molecules, 2025, doi:10.3390/molecules30173456_

Round 1

Reviewer 1 Report

Comments and Suggestions for Authors

Despite the need for certain corrections, I consider that the proposed manuscript meets the criteria to serve as a reliable reference source.

Author Response

Responses to Reviewer 1

The manuscript addresses a topic of considerable relevance and public health impact: the use of natural products to modulate inflammation, with a focused discussion on iridoid compounds. Iridoids are increasingly recognized for their anti-inflammatory potential through the several mechanisms and pathways involved. Their structural diversity and presence in medicinal plants make them valuable candidates for therapeutic development. By integrating a regional analysis of iridoid-producing plant species in country Bulgaria, the manuscript adds depth from a phytochemical and biodiversity perspective. This dual focus—mechanistic insight and botanical mapping—aligns well with Molecules’ scope in natural product chemistry and pharmacognosy. The study presented is therefore of clear scientific interest and contemporary significance. Nonetheless, in my view, certain corrections are necessary in order for the manuscript to achieve the level of impact that its premise suggests and deserves. Some of my suggestions are outlined below.

Comments:
Comment 1: Structure of the review

In my view, the structure of the review would benefit from a thoughtful reorganization. While previous reviews have addressed the anti-inflammatory activity of iridoids, I believe that any manuscript focusing on this topic should begin by outlining the fundamental structural features of iridoids and the principal mechanisms underlying their anti-inflammatory effects. This foundational context would provide a coherent framework before transitioning to the specific Bulgarian plant species that contain these compounds. Subsequently, it may be appropriate to discuss plants that are ethnobotanically recognized for their anti-inflammatory properties, and those for which such activity has been experimentally validated.
Furthermore, although the central aim of the manuscript is not to comprehensively review the antiinflammatory potential of iridoids, but rather to connect this activity with species native to Bulgaria, it makes virtually no reference to prior reviews that explore the relationship between iridoid content and anti-inflammatory action. Integrating and critically analyzing this existing literature would not only reinforce the scientific foundation of the article, but also contribute to its overall coherence and stylistic elegance.
Response: We thank the reviewer for their valuable suggestions. In the revised manuscript, the structure has been improved to address these concerns. The structural features of iridoids are now discussed in the context of their mechanistic behaviour in Section 3, while potential toxicity mechanisms are addressed in the newly added Section 4. We have also clearly separated the discussion of therapeutic effects and mechanisms of action of plants and their preparations and individual iridoids, acknowledging that specific iridoids (e.g., catalpol, aucubin) are not exclusive to the Bulgarian species covered.
Regarding the suggestion to include other plants ethnobotanically recognized for anti-inflammatory activity, we note that such a list would be practically infinite, as anti-inflammatory effects are common across a vast range of phytochemicals (e.g., polyphenols are ubiquitous in plants). This lies beyond the scope of our targeted focus on Bulgarian species rich in iridoids. Nonetheless, we have integrated and critically discussed relevant prior reviews on the relationship between iridoid content and anti-inflammatory action to strengthen the scientific foundation and coherence of the manuscript.

Comment 2: From my perspective, Table 1 currently detracts from the clarity of the manuscript rather than guiding the reader. It is densely populated with information that has not yet been logically introduced within the narrative, which may lead to confusion. Moreover, the table contains internal inconsistencies, introduces conceptual inaccuracies, and does not make appropriate use of the supporting literature. I outline my specific observations below to substantiate these concerns.
Response: In the revised manuscript, we have carefully addressed the noted discrepancies and restructured the table to improve clarity and logical flow. Specifically, all entries have been cross-checked for accuracy, internal consistency, and alignment with the narrative of the manuscript.

Comment 3: Table 1, conceptual Inaccuracies: Certain compounds listed—such as Plantamajoside and Acteoside—are not iridoids but phenylpropanoid glycosides. This misclassification is particularly problematic given that the manuscript itself correctly identifies Acteoside as a phenylpropanoid in lines 240–242.
Response: We thank the reviewer for highlighting this important point. In the revised version of the manuscript, both acteoside (verbascoside) and plantamajoside have been removed from the content of Table 1 to avoid misclassification. Although these phenylpropanoid glycosides share high structural and mechanistic similarities and often co-occur with iridoids in plants, they are chemically distinct and not members of the iridoid class. Their initial inclusion was intended to reflect their frequent co-presence in the studied species; however, their removal ensures complete alignment between the tabulated content and the chemical classifications described in the main text.

Comment 4: Table 1, Inconsistent Use of References: Several entries lack appropriate bibliographic support or cite sources that do not substantiate the claims made:
Response: Thank you for your remarks. In the revised manuscript, all entries in Table 1 have been cross-checked against the cited literature to ensure that each statement is directly supported by the appropriate bibliographic source. Where necessary, additional primary references have been incorporated.

Comment 5: Reference 9 (Veronica officinalis): Although Monotropein is indeed an iridoid compound, it has not been isolated from this species, nor from its genus or family. Associating this compound with Veronica officinalis is therefore unsupported and misleading.
Response: We have accordingly repositioned the citation source for monotropein.

Comment 6: Reference 11 is a review focused on verbascoside, a phenylpropanoid glycoside that is not structurally classified as an iridoid. While verbascoside is indeed biosynthesized in Plantago spp., its inclusion in a table specifically dedicated to iridoid compounds and their biological activity is conceptually inconsistent with the stated scope of the manuscript.
Response: Thank you for pointing out the discrepancy. Upon review, we determined that this misplacement resulted from a bibliographic software error, as the author (Alipieva) is the same for both the correct iridoid-related reference and the verbascoside-focused paper. The table has been corrected accordingly.

Comment 7: Reference 12: Similar to Reference 13, this citation refers to studies evaluating the antiinflammatory activity of plant extracts—often a diverse array of extract types—without assessing the presence or role of iridoids. As such, it does not substantiate a direct link between iridoid content and anti-inflammatory activity, and may not be representative for inclusion.

Comment 8: Reference 13: This is an original research article focused on the antimicrobial and antioxidant activity of Plantago spp., with no specific relevance to iridoid-mediated anti-inflammatory effects. The section on anti-inflammatory activity cites studies on total extracts rather than isolated iridoids, making it unsuitable for the current discussion.

Comment 9: Based on this analysis, the only references that could reasonably be retained in a revised version of Table 1—regarding Plantago species—are References 14–16, which address asperuloside. However, asperuloside is not the only iridoid compound reported for this genus; therefore, the table as presented does not provide a comprehensive overview of iridoid production within Plantago. Notably, a key reference (https://doi.org/10.1007/978-3-642-22144-6_295-1: Iridoid glycosides of Plantago: Botany, chemistry, bioactivity, and metabolism) has not been cited, despite its relevance. This omission further underscores the need to reassess the content and placement of Table 1. In my view, the table in its current form is more likely to mislead than to inform, and its exclusion should be considered.
Responses to Comments 7-9:
We thank the reviewer for their detailed evaluation and constructive feedback regarding the references and the content of Table 1.
Our original intention in compiling Table 1 was to provide a broader overview of the reported therapeutic effects associated with the selected species and their preparations, including activities such as anti-inflammatory, antimicrobial and antioxidant effects. However, in response to the reviewer’s request, we have reassessed and substantially revised the scope and structure of the table. It now focuses exclusively on the molecular mechanisms and therapeutic potential of specified iridoid compounds, rather than general extract-based studies or non-specific bioactivities. This shift in focus ensures that the table is better aligned with the core objectives of the manuscript and avoids misaligned whole-extract effects with those attributable to isolated iridoids.
Accordingly, References 12 and 13 have been removed, as they do not provide direct evidence for iridoid-mediated anti-inflammatory activity and are thus not suitable under the revised scope. We have retained only those references (now updated as per the revised Table 1) that specifically address the activity or presence of iridoid compounds.

Regarding the suggestion to cite the reference associated with the DOI https://doi.org/10.1007/978-3-642-22144-6_295-1 number, we attempted to access this source but could not verify its existence or relevance, as the DOI appears invalid or does not lead to a currently accessible or identifiable publication. We would appreciate clarification or an alternative reference if a specific publication was intended.
We believe that the revised version of Table 1 now more accurately reflects the state of knowledge on iridoid compounds and their therapeutic potential and avoids any misrepresentation.

Comment 10: Throughout the manuscript, the term native plants is used to describe species that, for the most part, are not originally from Bulgaria but are either cosmopolitan taxa or introduced species. It is recommended to replace this term with a more accurate expression, such as plants growing spontaneously in Bulgaria or species naturalized in Bulgaria. If the list includes species that are truly indigenous, the correct wording would be native or naturalized species in Bulgaria. This revision will improve the biogeographical and taxonomic accuracy of the text. See DOI: 10.1046/j.1472- 4642.2000.00083.x
Response: We respectfully note that, according to the World Flora Online database (https://www.worldfloraonline.org/), all plant species included in our study are recorded as having a native distribution in Bulgaria. They are not classified as cosmopolitan, introduced, or naturalized species. World Flora Online is a taxonomically authoritative resource developed by a consortium of botanical institutions to provide a consensus classification and verified distribution data for all known plant species. In light of this evidence, we have retained the term native plants in the manuscript, as it accurately reflects the current global taxonomic consensus for the studied species.

Comment 11: Notwithstanding the above observations, I consider that the proposed paper possesses sufficient merits to warrant publication once these minor corrections have been addressed.
Response: We thank the reviewer for their positive assessment and constructive feedback. We have accordingly addressed the suggested minor corrections throughout the manuscript. We hope that the revisions we provided meet your expectations and enhance the clarity and accuracy of the paper. We appreciate your support for publication and look forward to any further recommendations you may have.

Reviewer 2 Report

Comments and Suggestions for Authors

Comments:

   The manuscript entitled “Targeting inflammation with natural products: a mechanistic review of iridoids from Bulgarian medicinal plants”. Chronic, low-grade systemic inflammation is increasingly recognized as a core driver of the pathogenesis of numerous noncommunicable diseases (NCDs). The primary objective of this review is to analyze the anti-inflammatory activities of iridoid compounds from ten Bulgarian medicinal plants, with a focus on their interactions with key molecular targets and pathways involved in the inflammatory process. Furthermore, the available evidence regarding the in vitro and in vivo therapeutic potential of their preparations and major bioactive compounds is summarized, identifying their relevance as alternatives or complements to traditional anti-inflammatory therapies. However, several critical aspects warrant further clarification.

Comments:

  1. The report could be further streamlined to highlight the core findings and significance of the research in the abstract. For example, the key mechanisms and potential applications of iridoids in anti-inflammatory effects could be more clearly pointed out..

  1. Comparisons of iridoids with other natural anti-inflammatory compounds (e.g., flavonoids and polyphenols) could be expanded to highlight their unique properties..

  1. The main iridoid function should build the table.

  1. Although the article mentions some in vivo and in vitro studies, some data or prospects of clinical trials can be added to enhance the persuasiveness.

  1. The article mentions the high safety profile of iridoids, but further information on potential side effects or limitations could be added to provide a more comprehensive picture. For some mechanisms that have not yet been fully elucidated, future research directions could be identified.

Author Response

Responses to Reviewer 2

The manuscript entitled “Targeting inflammation with natural products: a mechanistic review of iridoids from Bulgarian medicinal plants”. Chronic, low-grade systemic inflammation is increasingly recognized as a core driver of the pathogenesis of numerous noncommunicable diseases (NCDs). The primary objective of this review is to analyze the anti-inflammatory activities of iridoid compounds from ten Bulgarian medicinal plants, with a focus on their interactions with key molecular targets and pathways involved in the inflammatory process. Furthermore, the available evidence regarding the in vitro and in vivo therapeutic potential of their preparations and major bioactive compounds is summarized, identifying their relevance as alternatives or complements to traditional anti-inflammatory therapies. However, several critical aspects warrant further clarification.

Comments:

Comment 1:  The report could be further streamlined to highlight the core findings and significance of the research in the abstract. For example, the key mechanisms and potential applications of iridoids in anti-inflammatory effects could be more clearly pointed out.
Response: Thank you for the constructive comment regarding the need to streamline the abstract and emphasize the key mechanisms and potential applications of iridoids in anti-inflammatory effects. We have revised the abstract to more clearly highlight the principal signaling pathways involved (NF-κB, MAPK, JAK/STAT, AMPK, and PPAR) and to summarize the broad therapeutic potential of iridoids, integrating both mechanistic insights and evidence-based pharmacological activities (lines 16-34)

Comment 2: Comparisons of iridoids with other natural anti-inflammatory compounds (e.g., flavonoids and polyphenols) could be expanded to highlight their unique properties.
Response: We appreciate the reviewer’s suggestion and have accordingly expanded the Introduction section to provide a clearer comparison between iridoids and other well-studied classes of natural anti-inflammatory compounds, particularly flavonoids and polyphenols. We also highlight unique mechanistic properties of iridoid compounds and potential synergistic interactions with polyphenolic compounds, underscoring their complementary modes of action and combined therapeutic potential.

Comment 3:  The main iridoid function should build the table.
Response: Thank you for your recommendation. As requested, Table 1 has been reconstructed, summarizing main iridoid molecular mechanisms and therapeutic effects.

Comment 4:  Although the article mentions some in vivo and in vitro studies, some data or prospects of clinical trials can be added to enhance the persuasiveness.
Response: We thank the reviewer for this valuable suggestion. In response, we have accordingly included the available clinical data highlighting the therapeutic potential of several iridoid compounds and preparations derived from their source plants. This addition provides evidence from human studies on safety, efficacy, and specific indications, thereby strengthening the translational relevance and persuasiveness of our article.

Comment 5:  The article mentions the high safety profile of iridoids, but further information on potential side effects or limitations could be added to provide a more comprehensive picture. For some mechanisms that have not yet been fully elucidated, future research directions could be identified.
Response: Thank you for your note. As recommended, we have included an additional section 4 in the revised manuscript titled “Current limitations and future perspectives for iridoid-based therapeutics”.

Reviewer 3 Report

Comments and Suggestions for Authors

Review of the manuscript

Journal

Molecules (ISSN 1420-3049)

Manuscript ID

molecules-3819898

Type

Review

Title

Targeting inflammation with natural products: a mechanistic review of iridoids from Bulgarian medicinal plants

Authors

Rositsa Mihaylova * , Viktoria Elincheva , Reneta Gevrenova , Dimitrina Zheleva-Dimitrova , Georgi Momekov , Rumyana Simeonova *

Section

Natural Products Chemistry

Special Issue

Role of Natural Products in Inflammation

_____________________________________________

Overall comments:

  • Based on the statement that chronic, low-grade systemic inflammation is increasingly recognized as a central driver in the pathogenesis of numerous non-communicable diseases and that conventional anti-inflammatory therapies, such as NSAIDs and corticosteroids, often present safety concerns with long-term use, medicinal plants have shown promise in modulating key inflammatory pathways through diverse mechanisms of action. The primary aim of this review is to provide a systematic and mechanistic summary of the anti-inflammatory activities of plant-derived iridoids, focusing on their interactions with key molecular targets and pathways involved in inflammatory processes.
  • Although the article points out the favorable safety profile of iridoids, there is no discussion of chronic toxicity, genotoxicity, or drug interactions.
  • Despite citing "safety profiles" in the text, safety is not efficiently assessed.
  • In lines 21-23 the authors point that “

The primary aim of this review is to provide a systematic and mechanistic summary of the anti-inflammatory activities of plant-derived iridoids, focusing on their interactions with key molecular targets and pathways involved in inflammatory processes.” I ask the authors: Can this review be considered a systematic review?

The term systematic review appears another 2 times in the text, such as in the Conclusion section (“This review presents a comprehensive and systematic overview of the anti-inflammatory potential of iridoid-containing medicinal plants native to the Bulgarian flora”.)
It is necessary to exercise caution to say “systematic” because this type of review needs many steps to be concluded:
a) Lack of Registered Protocol or Explicit Methodology:

-there is no description of search strategies (databases, MeSH terms, filtering criteria by date or language); Neither PRISMA flowchart showing the study screening process (identification, selection, eligibility, inclusion), nor are there specified Inclusion/Exclusion Criteria; There is not table showing the bias and study quality; Does not use validated tools (e.g., Cochrane ROB, SYRCLE for animals) to assess the risk of bias of the included studies.

________________________________

TITLE

        The title is adequate.

ABSTRACT

The abstract isn’t long. I suggest defining all the acronyms that appear. There’s space for that.

KEYWORDS

The Keywords are

 Keywords: iridoids, inflammation, metabolic syndrome, NFkB, JAK/STAT, antioxidant, 30 AMPK, catalpol, aucubin, NCDs

I suggest:

Keywords: iridoids, inflammation, antioxidant, metabolic syndrome, NFkB, JAK/STAT, AMPK, catalpol, aucubin, non-communicable diseases

INTRODUCTION

There is much new information that could be included in the Introduction section. I suggest the authors include more modern references that contextualize the topic. Are there conflicting information regarding the plants

RESULTS

  • Figure 1 is of poor quality. On the other hand, Figure 1 is beautiful. However, Figure 2, although beautiful, is very "polluted", making it difficult for the reader to understand.

CONCLUSION

      The conclusion is adequate.

I suggest including a separate section on the limitations of this study.

Please include a new section named Future Perspectives. In this section, for example, please explore the technological perspectives and benefits to future health medicine regarding iridoids.

REFERENCES

As mentioned above, please include newer references.

Author Response

Responses to Reviewer 3

Based on the statement that chronic, low-grade systemic inflammation is increasingly recognized as a central driver in the pathogenesis of numerous non-communicable diseases and that conventional anti-inflammatory therapies, such as NSAIDs and corticosteroids, often present safety concerns with long-term use, medicinal plants have shown promise in modulating key inflammatory pathways through diverse mechanisms of action. The primary aim of this review is to provide a systematic and mechanistic summary of the anti-inflammatory activities of plant-derived iridoids, focusing on their interactions with key molecular targets and pathways involved in inflammatory processes.

Comments:

Comment 1: Although the article points out the favorable safety profile of iridoids, there is no discussion of chronic toxicity, genotoxicity, or drug interactions. Despite citing "safety profiles" in the text, safety is not efficiently assessed
Response: Thank you for your constructive remarks. In response, we have added a new Section 4, titled “Current Limitations and Future Perspectives for Iridoid-Based Therapeutics,” which addresses these issues in detail. This section discusses the current limitations in understanding chronic toxicity, genotoxicity, and potential drug interactions, providing a more comprehensive evaluation of the safety profile of iridoid compounds and highlighting areas for future research.

Comment 2: In lines 21-23 the authors point that “The primary aim of this review is to provide a systematic and mechanistic summary of the anti-inflammatory activities of plant-derived iridoids, focusing on their interactions with key molecular targets and pathways involved in inflammatory processes.” I ask the authors: Can this review be considered a systematic review? The term systematic review appears another 2 times in the text, such as in the Conclusion section (“This review presents a comprehensive and systematic overview of the anti-inflammatory potential of iridoid-containing medicinal plants native to the Bulgarian flora”.) It is necessary to exercise caution to say “systematic” because this type of review needs many steps to be concluded:
a) Lack of Registered Protocol or Explicit Methodology:
-there is no description of search strategies (databases, MeSH terms, filtering criteria by date or language); Neither PRISMA flowchart showing the study screening process (identification, selection, eligibility, inclusion), nor are there specified Inclusion/Exclusion Criteria; There is not table showing the bias and study quality; Does not use validated tools (e.g., Cochrane ROB, SYRCLE for animals) to assess the risk of bias of the included studies.
Response: We thank the reviewer for this important observation. We agree that the term “systematic review” implies adherence to formal methodologies, including registered protocols, defined search strategies, inclusion/exclusion criteria, and risk-of-bias assessments, which were not applied in the current manuscript. Accordingly, we have removed the term “systematic” throughout the text and replaced it with “comprehensive” or “detailed”, which more accurately reflect the nature of the review as a thorough overview of the anti-inflammatory activities of plant-derived iridoids. This revision clarifies that the manuscript provides an in-depth summary and mechanistic insights without implying adherence to formal systematic review protocols.

Comment 3: TITLE. The title is adequate.
Response: Thank you for your positive feedback. We aimed to ensure the proposed title accurately reflects the content and focus of the manuscript.

Comment 4: ABSTRACT. The abstract isn’t long. I suggest defining all the acronyms that appear. There’s space for that.
Response: Thank you for your helpful suggestion. We have revised the abstract to define all acronyms upon first use, as recommended. The abstract remains within the allowable length and is now more accessible to readers who may be unfamiliar with the terminology.

 Comment 5: KEYWORDS. The Keywords are:
Keywords: iridoids, inflammation, metabolic syndrome, NFkB, JAK/STAT, antioxidant, 30 AMPK, catalpol, aucubin, NCDs
I suggest:
Keywords: iridoids, inflammation, antioxidant, metabolic syndrome, NFkB, JAK/STAT, AMPK, catalpol, aucubin, non-communicable diseases
Response: Thank you for your suggestion regarding the listed keywords. We have accordingly rearranged them and replaced the abbreviation "NCDs" with the full term "non-communicable diseases" to enhance clarity and accessibility.

 Comment 6: INTRODUCTION. There is much new information that could be included in the Introduction section. I suggest the authors include more modern references that contextualize the topic. Are there conflicting information regarding the plants
Response: Thank you for your suggestion. We have revised the Introduction to incorporate more recent and relevant references to better contextualize the topic. Additionally, we have included a brief discussion of conflicting findings related to the studied plants where applicable (In the newly added Section 4.1.), to provide a more balanced and comprehensive overview of the current state of knowledge:
“Notably, both pro-apoptotic and cytoprotective outcomes have been reported for certain iridoid-rich species, depending on context. For example, Plantago major leaf extracts (rich in aucubin) show selective cytotoxicity against certain breast cancer cells, yet aucubin alone was non-toxic except at very high concentrations and robustly suppressed neutrophil ROS production. https://doi.org/10.1186/s43046-019-0010-3; doi: 10.4103/pm.pm_406_1. Meanwhile, verbascoside (from Verbascum spp.) induced apoptosis in colorectal and breast cancer cells via p53 activation but remains safe for normal cells. https://doi.org/10.1186/1471-2407-14-747;  doi: 10.1186/s40360-021-00540-8. These findings align with the concept that iridoids exhibit bias effects on cell proliferation, depending on the experimental conditions (e.g. physiological or pathological settings), preserving healthy tissues through anti-inflammatory and anti-apoptotic pathways (e.g., NF-κB inhibition), while inducing apoptosis selectively in malignantly transformed cells.”

 Comment 7: RESULTS. Figure 1 is of poor quality. On the other hand, Figure 1 is beautiful. However, Figure 2, although beautiful, is very "polluted", making it difficult for the reader to understand.
Response: Thank you for your feedback on the quality of Figures 1 and 2. In the revised version of the manuscript, we have replaced Figure 1 with a higher-resolution version to ensure clarity and visual appeal. Regarding Figure 2, we have simplified the illustrated mechanistic pathways to reduce visual clutter and improve labeling readability, while retaining the essential information.

Comment 8: CONCLUSION. The conclusion is adequate. I suggest including a separate section on the limitations of this study. Please include a new section named Future Perspectives. In this section, for example, please explore the technological perspectives and benefits to future health medicine regarding iridoids.
Response: Thank you for your constructive suggestion. As recommended, we have included an additional section (Section 4) in the revised manuscript titled “Current Limitations and Future Perspectives for Iridoid-Based Therapeutics” This section outlines the safety concerns and pharmacokinetic limitations of iridoid compounds, along with innovative chemical and technological approaches to address them. We believe this addition strengthens the manuscript by offering a more forward-looking view of the topic.

 Comment 9: REFERENCES. As mentioned above, please include newer references.
Response: Thank you for pointing this out. We have reviewed and updated the reference list to include several more recent and relevant studies that better reflect the current state of research in this field.

Round 2

Reviewer 2 Report

Comments and Suggestions for Authors

Accept in present form

Reviewer 3 Report

Comments and Suggestions for Authors

Dear authors, 
Thank you for performing the corrections I suggested.
With best regards,
Dr Sandra M. Barbalho